# Large exchange-driven intrinsic circular dichroism of a chiral 2D hybrid perovskite

Shunran Li [1,2,9], Xian Xu [2,3,9], Conrad A. Kocoj [1,2], Chenyu Zhou [4], Yanyan Li[1,2], Du Chen [1,2], Joseph A. Bennett[2,5], Sunhao Liu[6], Lina Quan [6,7], Suchismita Sarker [8], Mingzhao Liu [4], Diana Y. Qiu [2,3] ✉ & Peijun Guo [1,2] ✉

In two-dimensional chiral metal-halide perovskites, chiral organic spacers endow structural and optical chirality to the metal-halide sublattice, enabling exquisite control of light, charge, and electron spin. The chiroptical properties of metal-halide perovskites have been measured by transmissive circular dichroism spectroscopy, which necessitates thin-film samples. Here, by developing a reflection-based approach, we characterize the intrinsic, circular polarization-dependent complex refractive index for a prototypical two-dimensional chiral lead-bromide perovskite and report large circular dichroism for single crystals. Comparison with ab initio theory reveals the large circular dichroism arises from the inorganic sublattice rather than the chiral ligand and is an excitonic phenomenon driven by electron-hole exchange interactions, which breaks the degeneracy of transitions between Rashba-Dresselhaus-split bands, resulting in a Cotton effect. Our study suggests that previous data for spin-coated films largely underestimate the optical chirality and provides quantitative insights into the intrinsic optical properties of chiral perovskites for chiroptical and spintronic applications.

Two-dimensional (2D) metal-halide perovskites (MHPs) are solution-processable, chemically diverse semiconductors comprised of inter-penetrated organic and inorganic sub-lattices[1–3]. The organic spacer cations, which impose a strong confinement on the charge carriers, give rise to a large exciton binding energy[4–6], tunable bandgaps[7], adjustable cross-layer energetic landscapes[8], and enhanced material stability[9]. The incorporation of chiral organic spacers into 2D and lower-dimensional MHPs offers a unique opportunity for materials design since the chirality of the organic spacers can be imparted onto the charge-residing inorganic sublattices, modifying the electronic and optical properties of these diverse hybrid semiconducting materials[10]. The interplay among polarization of light, spin of charge carriers, and structural chirality of these crystalline solids have enabled exciting properties including circularly polarized light emission and

detection[11–13], as well as spin-polarized charge transport originating from the chiral-induced spin selectivity (CISS) effect[14,15]. Recently emerged interesting phenomena in chiral MHPs include chiral phonon-induced spin currents[16], bulk photovoltaic effects[17], spin-dependent photogalvanic response[18], as well as reversible phase change between crystalline and amorphous states[19].

Circular dichroism (CD) spectroscopy has been the workhorse technique for understanding chiral molecules and solids, including chiral MHPs[20,21]. As a well-established optical technique, CD spectroscopy is conducted in a transmission configuration with samples taking the form of liquids (for molecules and nanocrystals) or films (for solids). The CD is defined on the basis of differential absorbance of left-handed circularly polarized (LCP) light and right-handed circularly polarized (RCP) light, usually expressed by the ellipticity ($\theta$) in

[1]Department of Chemical and Environmental Engineering, Yale University, New Haven, CT, USA. [2]Energy Sciences Institute, Yale University, West Haven, CT, USA. [3]Department of Mechanical Engineering and Materials Science, Yale University, New Haven, CT, USA. [4]Center for Functional Nanomaterials, Brookhaven National Laboratory, Upton, NY, USA. [5]Department of Chemistry, Yale University, New Haven, CT, USA. [6]Department of Chemistry, Virginia Tech, Blacksburg, VA, USA. [7]Department of Materials Science and Engineering, Virginia Tech, Blacksburg, VA, USA. [8]Cornell High Energy Synchrotron Source, Cornell University, Ithaca, NY, USA. [9]These authors contributed equally: Shunran Li, Xian Xu. ✉e-mail: diana.qiu@yale.edu; peijun.guo@yale.edu

millidegrees (mdeg). To yield good signal contrast, a sample for CD measurement needs to exhibit intermediate optical density (OD), and the $\theta$ is dependent on several extrinsic parameters including the concentration of chiral species, optical interaction length, choice of substrate, and statistics of domain orientations[22,23].

Normalizing the $\theta$ by a sample's OD can yield a concentration- or thickness-independent anisotropy factor

$$g_{CD} = \frac{CD\,(mdeg)}{32980 \times OD} \tag{1}$$

Nevertheless, neither CD nor $g_{CD}$ from CD spectroscopy contains complete information on the material's intrinsic chiroptical properties. Indeed, the most fundamental linear optical property for chiral 2D-MHPs, as optically absorbing materials, is the circular polarization (CP)-dependent, complex refractive index (RI). Specifically, with RI written as $n + ik$ where $n$ ($k$) is the real (imaginary) part of RI, CD spectroscopy does not inform separately on $n$ and $k$, despite the influence of both $n$ and $k$ on the measured transmittance and the transmittance-derived CD. Furthermore, due to optical opaqueness in the above-bandgap range, the transmission configuration is incompatible for measuring MHP single crystals, which are free from defects and grain-to-grain variations present in thin-film samples. Resultingly, literature-reported CD spectra for chiral MHPs thus far are based on spin-coated thin films and hence likely represent properties averaged over grains and domains with different crystalline orientations.

Here, we report on a reflection-based technique and extract the CP-dependent RI for single crystals of a model chiral 2D-MHP, (S-NEA)$_2$PbBr$_4$ (denoted as S-NPB) and (R-NEA)$_2$PbBr$_4$ (R-NPB), where NEA$^+$ stands for 1-(1-naphthyl)ethylammonium cation[24]. We determine the CP-dependent RI in the optically absorbing spectral range of the two enantiomers and compare how they behave differently for LCP and RCP light. Importantly, we show that thin film-based CD measurements largely underestimate the intrinsic chiroptical behavior, likely due to domain variations, poor crystallinity, and defects. The measurements agree with our first-principles calculations based on the GW plus Bethe Salpeter equation (GW-BSE) approach within many-body perturbation theory. The calculations reveal that the observed exciton band-edge CD arises from excitons residing primarily in the inorganic sublattice and is driven by a purely quantum mechanical electron-hole exchange interaction, which lifts the degeneracy of degenerate bright exciton states arising from transitions between the Rashba-Dresselhaus (RD)-split bands. Our approaches can be generalized to accurately characterize and engineer the chiroptical properties of other extended systems employing chiral ligands.

## Results

### Determination of the complex refractive indices

The optical and scanning electron micrographs, and the X-ray diffraction pattern (Supplementary Fig. 1), reveal the high quality of centimeter-sized single crystals exhibiting optically smooth a-b facets parallel to the lead-bromide layers, allowing us to access the intrinsic optical properties of these materials. The optical measurements were performed with light perpendicularly incident on the a-b plane of the crystals. Since S/R-NPB with a layered structure has drastically different optical responses along the in-plane and out-of-plane directions[25], the wave vector of light used throughout our measurements dictates that the optical properties discussed here are exclusively in-plane. Supplementary Fig. 2 depicts the experimental setup developed for the work, with samples measured in reflection (see Methods for experimental details). A photoelastic modulator (PEM) was used to periodically alternate the incident light between LCP and RCP at the PEM's fundamental frequency of ~50 kHz, which we denote as the 1$f$ frequency.

Before addressing the CP-dependent RI, we first work out the RI under unpolarized light by combining micro-reflectance spectroscopy and a dielectric coating-based method[25]. As a model-blind approach that does not require parametric fitting, the method involves two separate reflectivity measurements: the first measurement is on a pristine perovskite crystal, and the second is on a perovskite crystal coated with a thin dielectric layer (with known thickness and RI). As reflectivity depends on the perovskite's $n$ and $k$[26], using the two measured reflectivity values as independent input parameters, the perovskite's complex RI (i.e., both $n$ and $k$) can be determined without invoking any fitting parameters (see Supplementary Note 1 for details). The reflection-based method is particularly suited for the spectral range where the material is a strong light absorber, as light is completely terminated within the crystal with absence of backside reflection off of the crystal. For this reason, the long wavelength limit of this work is ~402 nm, dictated by the material bandgap. The short wavelength is cut off at 365 nm, determined by the available light source in this work. As discussed below, the 365 - 402 nm range completely covers the exciton resonance of S/R-NPB.

The dielectric coating we adopted is a thin Al$_2$O$_3$ layer grown by electron-beam evaporation. The RI and thickness ( ~ 17 nm) of Al$_2$O$_3$ are determined by spectroscopic ellipsometry (Supplementary Fig. 3). The perovskite's RI is then determined wavelength-by-wavelength. For instance, Fig. 1a, b demonstrate how the RI is determined for 386 nm. Using the transfer-matrix method or Fresnel's law of reflection (Supplementary Note 1), we first search for combinations of perovskite's $n$ and $k$ that are necessary to yield experimental reflectivity (15.3%) at this wavelength; these combinations are shown as a contour in Fig. 1a. We then determine what combinations of $n$ and $k$ must also be true, such that the Al$_2$O$_3$/perovskite stack reproduces the measured reflectivity (5.1%); these combinations appear as another contour in Fig. 1b. The intersection of these two contours uniquely determines the true $n$ and $k$ at 386 nm.

Figure 1c presents the reflectance spectra of S-NPB and R-NPB single crystals, measured with unpolarized light, both before and after coating by Al$_2$O$_3$. By executing the procedure outlined above, we determine the wavelength-dependent $n$ and $k$ for the two enantiomers from 365 to 402 nm (Fig. 1d). The slight difference between the two enantiomers likely arises from sample-quality variations and experimental errors, as we do not anticipate the two enantiomers to differ under unpolarized light. Notably, Fig. 1d reveals an absorption centered at 387 nm, which corresponds to the exciton resonance of S/R-NPB and is on par with exciton wavelengths of other $N = 1$ lead-bromide perovskites ($N$ denotes the number of octahedral layers per repeating unit along the c axis)[27]. Compared to (BA)$_2$PbI$_4$[25], an $N = 1$ lead-iodide perovskite where BA$^+$ = CH$_3$(CH$_2$)$_3$NH$_3^+$, the peak value of $k$ for S/R-NPB at the exciton resonance is lower (i.e., $k = 0.8$ for S/R-NPB versus $k = 1.5$ for (BA)$_2$PbI$_4$), which we attribute to the larger out-of-plane lattice constant of S/R-NPB that in turn reduces the exciton oscillator strength. The weaker exciton resonance is also manifested as a weaker exciton-induced reflectivity peak ( < 20%; Fig. 1c, before Al$_2$O$_3$ coating) compared to that of (BA)$_2$PbI$_4$ (>30%).

The $n$ and $k$ in Fig. 1d represent the response of S/R-NPB evenly averaged between LCP and RCP light with random phase delays. Equivalently, the reflectivity in Fig. 1c is equal to $\left( |r_L|^2 + |r_R|^2 \right)/2$, where $|r_L|^2 = R_L$ and $|r_R|^2 = R_R$ are the reflectivity for LCP light and RCP light, respectively. More information is still needed to determine the relative amplitude of $R_L$ and $R_R$, and with them the CP-dependent RIs. As discussed in Supplementary Note 2, the ratio of $R_R$ and $R_L$ is derived to be

$$R_R/R_L = \frac{2}{1 \pm 0.882\frac{I_{1f}}{I_{DC}}} - 1 \tag{2}$$

(the sign "$\pm$" takes "$-$" if $R_R - R_L > 0$, and "$+$" if $R_R - R_L < 0$), where $I_{1f}$ and $I_{DC}$ represent the intensities of reflected light modulated at the $1f$ frequency and the chopper frequency, respectively. Therefore, since $(|r_L|^2 + |r_R|^2)/2$ is already known, measuring the ratio $\frac{I_{1f}}{I_{DC}}$ will permit the quantification of $R_R$ and $R_L$ individually. Figure 2a presents the measured $\frac{I_{1f}}{I_{DC}}$ for $S$-NPB and $R$-NPB crystals, both before and after coating with $Al_2O_3$. In addition to the amplitudes, we find that the $1f$ signal for $R$-NPB and $S$-NPB exhibit phases offset by 180° on the lock-in amplifier, and the same phase offset persists throughout the 365 ~ 402 nm range, independent of the coating of $Al_2O_3$. Such a persistent phase difference arises from the opposite ranking of $R_R$ and $R_L$ between $R$-NPB and $S$-NPB over the explored spectral range: for $S$-NPB, we find $R_L < R_R$ (as depicted in Fig. 2b). It is important to note handedness inversion for CP light upon reflection (Supplementary Note 2), as illustrated in Fig. 2b. Note that we did not observe noticeable lock-in voltage at the $1f$ frequency for racemic-NPB crystals, suggesting its lack of CD response.

## Circular polarization-dependent optical responses

Combining $R_L + R_R$ and $R_R/R_L$, we can now determine the wavelength-dependent $R_L$ and $R_R$ individually, both before and after coating with $Al_2O_3$. These reflectivity values (specifically, four values at each wavelength for each enantiomer) finally permit extraction of the CP-dependent $n$ and $k$ for the two enantiomers. The complex RIs for $S$-NPB, including $n_{S-NPB,R}$, $n_{S-NPB,L}$, $k_{S-NPB,R}$ and $k_{S-NPB,L}$ (the second subscript, R or L, indicates RCP or LCP, respectively), are plotted in Fig. 2c. These data then permit the calculation of differences in the $n$ and $k$ between LCP and RCP for $S$-NPB (Fig. 2d). The results for $R$-NPB are summarized in Supplementary Fig. 4. Overall, the two enantiomers exhibit opposing chiroptical behaviors, *i.e.*, RCP light and LCP light for $R$-NPB are respectively as LCP light and RCP light for $S$-NPB. The slight differences in the amplitudes of $|\Delta n|$ and $|\Delta k|$ between the two enantiomers (Fig. 2d and Supplementary Fig. 4b) are likely not intrinsic but arise from experimental errors and sample-to-sample variations.

Using the extracted $n$ and $k$, we can further calculate the real and imaginary parts of the relative permittivity through $\varepsilon' = n^2 - k^2$ and $\varepsilon'' = 2nk$, respectively. These results, and their differences between LCP and RCP, are presented in Fig. 2e, f (Supplementary Figs. 4c, d) for $S$-NPB ($R$-NPB). The results for $S$-NPB (Fig. 2c–f) distill the following observations. Firstly, the shapes of $\varepsilon'$ and $\varepsilon''$, and their relative amplitudes between LCP and RCP (Fig. 2e), resemble those seen in $n$ and $k$ (Fig. 2c). Secondly, the $(k_L - k_R)_{S-NPB}$ curve exhibits a derivative-like lineshape (Fig. 2d), whose zero-crossing wavelength matches the exciton peak (Fig. 1d); such a derivative-like lineshape indicates a spectral blueshift of $k_{S-NPB}$ as light is switched from LCP to RCP (Fig. 2c) associated with CD. Thirdly, a negative dip is present in the spectra of $(\varepsilon'_L - \varepsilon'_R)_{S-NPB}$ and $(n_L - n_R)_{S-NPB}$; these dips reside on a dispersion-less negative background. Last but not least, a circular birefringence (CB), signified by $|(n_L - n_R)_{S-NPB}|$, is observed, which is on the order of 0.02–0.04. Notably, this exciton resonance-enhanced

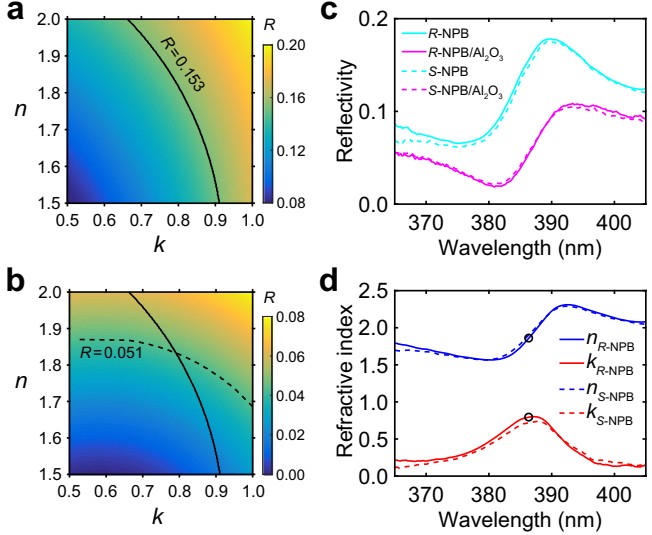

**Fig. 1 | Optical characterization of $S$-NPB single crystals for unpolarized light.** **a** Calculated reflectivity ($R$) of a bulk, light-absorbing material as a function of its $n$ and $k$. The black solid line traces the contour with a constant reflectivity of 0.153 (which corresponds to the reflectivity for $R$-NPB at 386 nm). **b** Calculated reflectivity of a bulk, light-absorbing material coated with a 17-nm thick layer of $Al_2O_3$ (refractive index shown in Supplementary Fig. 2) as a function of the $n$ and $k$ of the light-absorbing material. The black dashed line traces a contour with a constant reflectivity of 0.051 (which corresponds to the reflectivity for $R$-NPB coated with 17-nm thick $Al_2O_3$ at 386 nm). **c** Experimental reflectivities for $S$-NPB and $R$-NPB as a function of wavelength (cyan: uncoated; magenta: coated with 17-nm thick $Al_2O_3$). **d** Wavelength-dependent $n$ and $k$ values for $S$-NPB and $R$-NPB under un-polarized light determined using the dielectric-coating approach; the black circles indicate the wavelength of 386 nm, which was described in **a**, **b**.

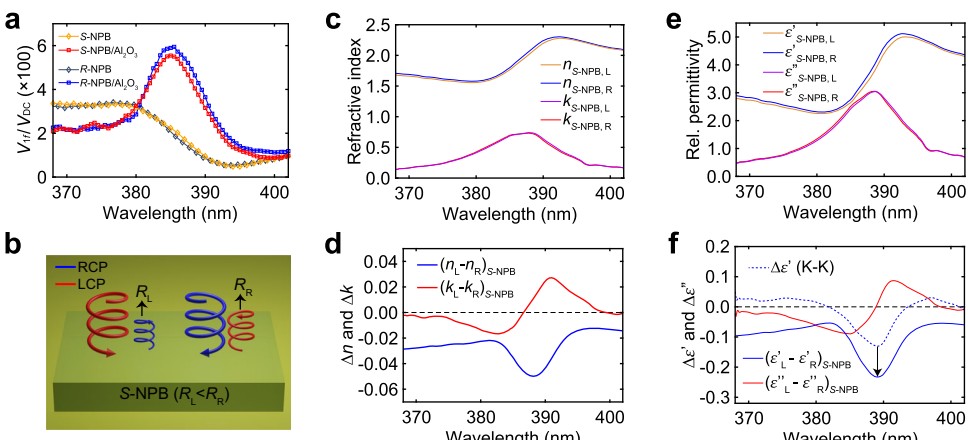

**Fig. 2 | Circular polarization-resolved optical characterization of $S$-NPB. a** Ratios of the voltages of the $1f$ and the DC components ($V_{1f}/V_{DC}$) as a function of wavelength measured with a lock-in amplifier; data are plotted for both $S$-NPB and $R$-NPB, and for both before and after the coating of $Al_2O_3$. **b** Schematic illustration of the LCP light and RCP light reflected off of an $S$-NPB single crystal ($R_L$: reflectivity for LCP light; $R_R$: reflectivity for RCP light). **c** CP-differentiated $n$ and $k$ for $S$-NPB. **d** Differences in $n$ and $k$ between LCP and RCP for $S$-NPB. **e** CP-differentiated $\varepsilon'$ and $\varepsilon''$ for $S$-NPB. **f** Solid lines: differences in $\varepsilon'$ and $\varepsilon''$ between LCP and RCP for $S$-NPB. Blue-dashed line: the change in $\varepsilon'$ caused by the change in $\varepsilon''$ calculated using the Kramers-Kronig (K-K) relations. RCP: right-hand circularly polarized light. LCP: left-hand circularly polarized light.

CB is 2–3 orders of magnitude higher than that of, *e.g.*, quartz, a prototypical circularly birefringent material, whose $|n_L − n_R| = 7.1 \times 10^{-5}$ at 589 nm[28]. The CD represented by $|(k_L − k_R)_{S-NPB}|$ has a magnitude of 0.02–0.04, which is comparable to the CB.

Using the Kramers-Kronig (K-K) relationship, the change in $\varepsilon'$ caused by a change in $\varepsilon''$ can be calculated as

$$\Delta\varepsilon'_{K-K}(\omega) = \frac{2}{\pi}\mathbf{P}\int_0^\infty d\omega_0 \frac{\omega_0 \cdot \Delta\varepsilon''(\omega)}{\omega_0^2 - \omega^2} \qquad (3)$$

with $\mathbf{P}$ denoting the principal value of the integral. The contribution to $(\varepsilon'_L − \varepsilon'_R)_{S-NPB}$ from $(\varepsilon''_L − \varepsilon''_R)_{S-NPB}$ obtained from the K-K analysis is shown in Fig. 2f as the blue-dashed line. Notably, the K-K analysis quantitatively captures the dip observed in the $(\varepsilon'_L − \varepsilon'_R)_{S-NPB}$ spectrum, leaving behind a nearly constant, negative residual of approximately −0.1, indicated by the arrow in Fig. 2f. Equivalently, $(\varepsilon'_L − \varepsilon'_R)_{S-NPB}$ is a superposition of $\Delta\varepsilon'_{K-K}$ (associated with $\Delta\varepsilon''$, *i.e.*, the CD effect) and a dispersion-less, negative shift of $\varepsilon'$ when going from LCP to RCP; the latter, negative component of $(\varepsilon'_L − \varepsilon'_R)_{S-NPB}$ indicates that $\varepsilon'_L < \varepsilon'_R$ for *S*-NPB. We assign this dispersion-less behavior of $\varepsilon'_L < \varepsilon'_R$ to the CB effect, which is complementary, and of similar magnitude, to the CD. The CB of *S*-NPB is consistent in sign with laevorotatory *S*-(-)-NEA molecules: they both exhibit $\varepsilon'_L < \varepsilon'_R$ or, equivalently, $n_L < n_R$, hence they both rotate the polarization of linearly polarized light counter-clockwise when viewed towards the light source. In contrast, *R*-NPB crystals exhibit an opposing behavior, *i.e.*, $\varepsilon'_L > \varepsilon'_R$ (Supplementary Fig. 4d), in line with the dextrorotatory response of *R*-(+)-NEA molecules.

Molar concentration-dependent polarimetry experiments on *R*-NEA and *S*-NEA molecules (Supplementary Fig. 5) show that in the solution phase, the optical rotation power is ±1.15 degrees cm$^{-1}$ (+ for *R*-NEA; − for *S*-NEA) at a concentration of 1 mol L$^{-1}$, translating to an amplitude of $|n_L − n_R| = 3.7 \times 10^{-7}$ L·mol$^{-1}$ or $|\varepsilon'_L − \varepsilon'_R| = 1.2 \times 10^{-6}$ L·mol$^{-1}$ for the chiral molecules (noting $n = 1.623$ for NEA at 589 nm). Based on the crystal structure of *S/R*-NPB, the equivalent molar concentration of NEA cations in *S/R*-NPB crystals is 4.9 mol L$^{-1}$. Interestingly, the CB exhibited by the perovskite solid is four orders of magnitude larger than that obtained by scaling the $|\varepsilon'_L − \varepsilon'_R|$ of *S/R*-NEA molecules to a concentration of 4.9 mol L$^{-1}$, suggesting the important role of the lead-bromide framework in the observed large CB effect.

## Ab initio many-body calculations

We then turn to theory to better understand the role of the lead-bromide framework in the large CD. We start by relaxing the experimental structure from literature[24] within density functional theory (DFT)[29]. We use the DFT ground-state calculated within the generalized gradient approximation of Perdew, Burke, and Ernzerhoff (PBE)[30] as a starting point for a one-shot $G_0W_0$ calculation to obtain the quasiparticle bandgap, followed by a GW-BSE calculation to obtain exciton eigenstates[31–33]. We then use Fermi's golden rule to calculate the contribution of the transition rate between the ground state and the exciton states to the $\varepsilon''$ associated with absorption of LCP and RCP light. The resulting spectra are shifted by 0.25 eV to compensate for underestimation of the bandgap resulting from the PBE ground-state, consistent with previous work[6,34–39]. Note that perturbations due to the electric dipole alone are not sufficient to describe CD, which also includes contributions from the magnetic dipole and, potentially, the electric quadrupole[40–45]. In our calculations, we included contributions from the electric dipole and the orbital component of magnetic dipole, which we find sufficient to understand the key features of the experimental results. Further computational details are in Supplementary Note 3 and Supplementary Figs. 6–8.

Figure 3a shows the theoretically calculated $\varepsilon''$ (and $\varepsilon'$ obtained from the K-K relation) overlaid on the experimental curves, and Fig. 3b

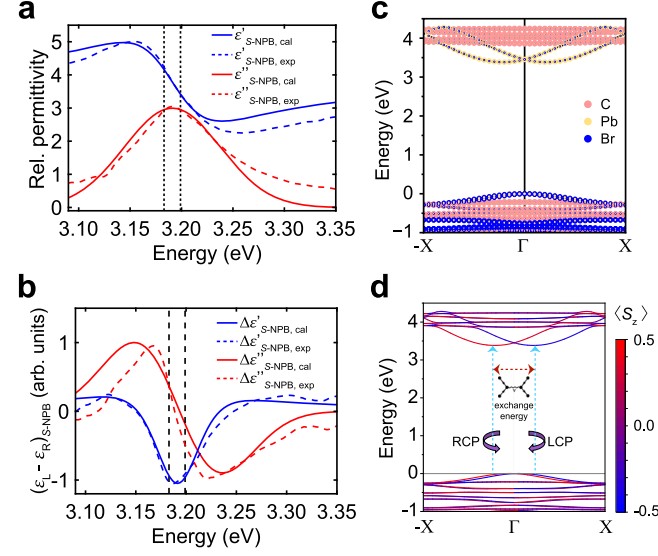

**Fig. 3 | First-principles calculations of CD of *S*-NBP. a** Comparison of theory (solid lines) and experiment (dashed lines) of the imaginary (blue) and real (red) parts of the relative permittivity of *S*-NPB. The theoretical peak has been shifted by 0.25 eV to match experiments, and a constant broadening of 46 meV has been used. **b** Differences, in arbitrary units, in the real and imaginary parts of the relative permittivity for LCP and RCP for *S*-NPB. **c** DFT band structure with $G_0W_0$ bandgap obtained from a scissors correction; the band structure is colored according to the orbital contribution from different atoms, and, **d**, the same band structure colored according to the projected component of the spin along the *z* direction, denoted as $\langle S_z \rangle$, with a schematic of the exchange interaction between electron–hole pairs on the different RD-split bands. RCP: right-hand circularly polarized light. LCP: left-hand circularly polarized light.

shows $\Delta\varepsilon''$ and $\Delta\varepsilon'$ between LCP and RCP, all for *S*-NBP. With a constant broadening of 46 meV, the theoretical results are in excellent agreement with experimental spectral features, including the derivative-like lineshape and sign changes in the $\Delta\varepsilon''$ spectrum. The binding energy of the lowest energy bright exciton of 0.46 eV is consistent with previously reported exciton binding energies in layered perovskites[46]. Importantly, we find that the lowest energy peak in $\varepsilon''$ consists of two distinct bright exciton resonances that are split by 17 meV. The lower-energy state is bright when the light is polarized along the Γ to X direction, whereas the higher-energy state is bright when the light is polarized along the Γ to Y direction (see Supplementary Fig. 7). Both states are bright under excitation with circularly polarized light. These two bright excitons are composed primarily of electrons in the two lowest conduction bands and holes in the two highest valence bands (see Supplementary Fig. 8f). Atomic-orbital-resolved band structures (Fig. 3c) and spin-resolved band structures (Fig. 3d) reveal that the highest valence bands and lowest conduction bands are RD-split states resulting from the Pb-Br octahedral tilting coupled with the large spin-orbit coupling (SOC) of Pb.

In an independent-particle picture, we would expect such states to give rise to two degenerate band-to-band transitions, one of which is allowed for RCP, and the other allowed for LCP due to selection rules imposed by the spin texture. Due to the degeneracy of these two sets of transitions, contrary to previous hypotheses that the RD splitting alone is responsible for chirality transfer between sublattices[47], we would not expect to observe CD (though CPL is still possible). However, the degeneracy of these transitions is broken after accounting for strong exciton effects in 2D perovskites[48]. As established in previous work[6,49], the undistorted layered perovskite has two degenerate low-energy spin-allowed exciton states that are bright under excitation with in-plane-polarized light. The octahedral tilting breaks the degeneracy of these bright excitons, and the two states exhibit a linear

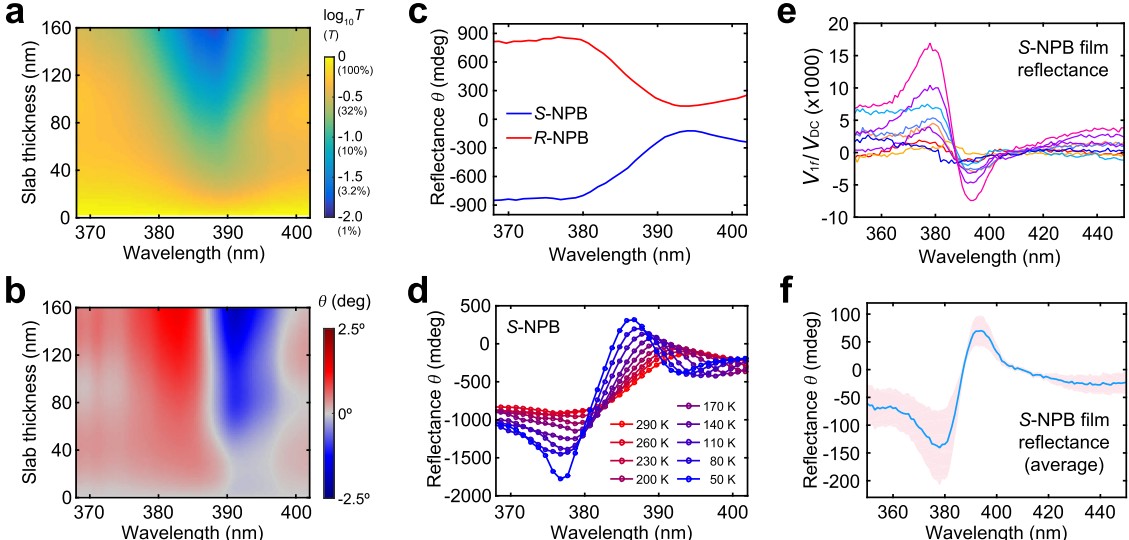

**Fig. 4 | Comparison of CD between single crystals and thin films. a** Optical transmittance (averaged between LCP and RCP) for an *S*-NPB single-crystalline slab as a function of slab thickness and wavelength, calculated using the refractive index data in Fig. 1d. The data is plotted in $log_{10}$ scale (the numbers in parentheses of the color bar show the transmittance values in a linear scale). **b** Map of the ellipticity $\theta$ (in degrees) as a function of wavelength and thickness of the single-crystalline *S*-NPB slab, calculated using the refractive index data in Fig. 2c. **c** Ellipticity angle $\theta$ measured for light reflected off of the *S/R*-NPB single crystals (calculated using data shown in Fig. 2a). **d** Ellipticity angle $\theta$ for light reflected off of the *S*-NPB single crystal at various temperatures from 290 K to 50 K. **e** Ratios of the 1*f* voltage ($V_{1f}$) and DC voltage ($V_{DC}$) of the lock-in measured in reflectance for a 320-nm thick, annealed crystalline *S*-NPB film. The data were taken at randomly selected spots on the sample. **f** Plots of average and standard deviation of ellipticity angle $\theta$ in reflectance, calculated using the data shown in **e**. The shaded area in **f** represents the standard deviation of the data shown in **e**.

dichroism with one state being coupled to light polarized along X direction and the other being coupled to light polarized along the Y direction. In the absence of exchange interactions, these bright spin-allowed states are degenerate with their spin-forbidden counterparts, and the states themselves are nearly degenerate with a small crystal-field splitting of ~1 meV. However, this near degeneracy is lifted by electron-hole exchange interaction, which splits the two sets of bright and dark states by different amounts, resulting in a ~17 meV splitting of the two bright states (Supplementary Note 3). This splitting of the bright states, which couple dissimilarly to RCP and LCP light, gives rise to a pronounced derivative-like line-shape in the $\Delta\varepsilon''$ spectrum, reminiscent of the Cotton effect[50]. In the absence of exchange, the excitonic states are nearly degenerate and the CD coming from band-edge excitons vanishes (Supplementary Fig. 6). Thus, the large chirality transfer from chiral spacers to lead-bromide sublattice, and the observed Cotton effect, are quantum phenomena driven by the different exchange-scattering of the bright excitons, which is in turn related to the crystal-field splitting.

**Comparing single crystals with thin films**

Back to the experimental results, having the RI data (Fig. 2c), we can calculate the transmittance of RCP and LCP light for an *S*-NPB single-crystalline slab with thickness varied within 2–160 nm. Figure 4a shows the calculated transmittance averaged for LCP and RCP light with the exciton absorption centered at 387 nm. With the CP-dependent RI, we can further calculate the ellipticity of light exiting the sample as

$$\tan\theta = \frac{E_R - E_L}{E_R + E_L} = \frac{\sqrt{I_R} - \sqrt{I_L}}{\sqrt{I_R} + \sqrt{I_L}} \qquad (4)$$

where $E_R$ ($E_L$) and $I_R$ ($I_L$) are the field strength and intensity of transmitted RCP (LCP) light, respectively. The map of $\theta$ (Fig. 4b) exhibits a derivative-like line-shape with a zero-crossing wavelength overlapping with the exciton peak, resembling the characteristic Cotton effect[50]. Notably, large values of $\theta$ over $\pm 1$ degree can be achieved for films thicker than ~100 nm; these large values of $\theta$ occur near, but not

exactly at, the exciton absorption peak. Similar results calculated for *R*-NPB are shown in Supplementary Fig. 9.

We can further define an ellipticity $\theta$ measured in reflectance by using $R_R$ and $R_L$ to replace $I_R$ and $I_L$ in Eq. (4), i.e.,

$$\tan\theta = \frac{E_R - E_L}{E_R + E_L} = \frac{\sqrt{R_R} - \sqrt{R_L}}{\sqrt{R_R} + \sqrt{R_L}} = \frac{\sqrt{R_R/R_L} - 1}{\sqrt{R_R/R_L} + 1} \qquad (5)$$

The one-to-one correspondence between $R_R/R_L$ and $\frac{I_{1f}}{I_{DC}}$ indicates that $\tan\theta$ can be directly calculated using $\frac{I_{1f}}{I_{DC}}$. As shown in Fig. 4c, large values of $\theta$ in reflectance can be achieved, although not as high as that in transmittance, which arises from a shorter optical interaction length in reflection (Supplementary Fig. 10). The sign of $\theta$ in reflectance is opposite to that in transmittance, consistent with the CP inversion upon reflection. Lowering the temperature can lead to enhancement in $\frac{I_{1f}}{I_{DC}}$ (Supplementary Fig. 11) and a more significant difference between $R_R$ and $R_L$. Resultingly, a large magnitude of $\theta$ approaching $-2°$ can be achieved at 50 K near the exciton resonance (Fig. 4d).

We then compare our results for single crystals to that of thin films, since films have been the predominant sample type for CD characterization. While the literature does not always report the film thickness together with CD spectra, it is reasonable to assume the thickness to be larger than 100 nanometers. Compared to reported thin-film CD spectra, whose magnitude typically resides in the tens-of-milli-degree range[24], the CD based on single-crystal results shown in Fig. 4 is 1~2 orders of magnitude stronger. Such a large discrepancy observed between film- and single crystal-chiral MHPs leads us to hypothesize that the chiroptical properties of spin-coated films may be strongly impacted by grains, grain boundaries, defects, and amorphous domains due to suboptimal crystallization. To test this hypothesis, we fabricated *S*-NPB and *R*-NPB films using literature-reported procedures[51]. Notably, the X-ray diffraction data for the annealed, crystalline film (Supplementary Fig. 12) lacks the high-index out-of-plane peaks exhibited by crystals (Supplementary Fig. 1d), suggesting a loss of long-range order. The annealed films are examined by cross-sectional scanning-electron microscopy and traditional CD

transmittance experiments (Supplementary Figs. 13a, b), demonstrating a lower ellipticity by approximately an order of magnitude when compared to the predicted single-crystal values. CD reflectance experiments on these films using our setup yielded consistently lower $\frac{V_{1f}}{V_{DC}}$ and ellipticity $\theta$ measured in reflectance (Fig. 4e, f for $S$-NPB and Supplementary Fig. 13c, d for $R$-NPB) in comparison to single crystals (Figs. 2a and 4d). Note that, as illustrated in Fig. 4e, we find the film to be highly nonuniform and its $\frac{V_{1f}}{V_{DC}}$ value varies from spot to spot.

Indeed, it has been found that chiral spacers can strongly disrupt crystallization of 2D-MHPs and facilitate the stabilization of amorphous phase of $S/R$-NPB[19]. We hence hypothesize the seemingly crystalline $S$-NPB films with out-of-plane alignment on the substrate host sizeable fractions of non-crystalline regions. Additional synchrotron X-ray diffraction data (Supplementary Fig. 14) illustrates a strong contrast in the crystallinity between single crystals and annealed films. Combining our experimental and theoretical results, we finally rationalize the following three possible reasons behind the observed lower CD for annealed films compared to single crystals.

Firstly, the non-crystallized domains in the film sample do not contribute to the CD near the exciton resonance, since the amorphous and crystalline phases have spectrally isolated exciton resonances (Supplementary Fig. 12c). In addition, our ab initio theory suggests that the selection rules imposed by the spin texture and RD splitting underpin the large CD around the exciton resonance. As such, the Cotton effect is sensitive to the energy splitting of the two exciton states, which is in turn sensitive to the crystal structure. In conjunction with the first point, we anticipate that lowered crystallinity, high-density grain boundaries (*e.g.*, those between domains of randomly distributed in-plane orientations), and defects can significantly relax the selection rules, thereby reducing the CD of the films. Lastly, the synchrotron X-ray diffraction data (Supplementary Fig. 14) further reveals that the annealed film has a slightly different lattice constant ( ~ 0.4% contraction) along the $c$ direction (*i.e.*, perpendicular to the 2D perovskite layers) from that of single crystals. This may lead to lower amount of octahedral tilting distortion, and with it weaker chiroptical effects.

In summary, using CP-resolved micro-reflectance spectroscopy, we determine the intrinsic, CP-distinguished refractive indices and relative permittivities of $S/R$-NPB. The chiroptical behavior consists of a CP-dependent spectral shift of exciton absorption and a dispersion-less CB in line with the optical activity of the chiral organic cations. The CD increases with decreasing temperature and approaches or exceeds unity degree in $\theta$ defined for transmittance or reflectance. Importantly, bulk crystals exhibit much larger CD response than films, which we attribute to imperfections and incomplete crystallization of the latter. Theoretical calculations show that the Cotton effect seen in the CD spectrum and enhanced CD at the band edge are excitonic effects driven by an electron-hole exchange interaction involving RD-split states at the band edge, suggesting new pathways for tuning the CD response through the exciton binding energy, the magnitude of the exchange, and the degree of inversion symmetry-breaking. Our work offers insights on chiral matter-light interactions towards enhanced chiroptical control of light emission and detection. The intrinsic CD characteristics and the concomitant theoretical understanding can be valuable for evaluating and controlling the degree of spin-splitting in the growing chiral MHP family with 2D and lower dimensionalities for luminescence, photodetection, and spintronic applications[52–56].

## Methods

### Material synthesis and sample fabrication

(S)−1-(1-naphthyl)ethylamine (*S*-NEA), (*R*)−1-(1-naphthyl)ethylamine (*R*-NEA) and lead (II) bromide (PbBr$_2$, 98%) were purchased from TCI Chemicals. Hydrobromic acid (HBr, 48%) and dimethyl formamide (DMF, 99.8%) were obtained from Sigma-Aldrich. All chemicals were used as received. $S$-NPB and $R$-NPB single crystals were grown by the controlled cooling method[19]. Specifically, 180 mg PbBr$_2$ and 156 μL $S$-NEA or $R$-NEA were dissolved in 2 mL of HBr and 4.8 mL of H$_2$O in a 25 mL glass bottle, and the solution was heated to and maintained at 95 °C. After the solutes were completely dissolved, the temperature of the precursor was first cooled down to 55 °C at a rate of 5 °C h$^{-1}$, and then cooled down to room temperature at a rate of 0.5 °C h$^{-1}$. Large $S$-NPB and $R$-NPB single crystals slowly formed during the second cooling process. The crystals were harvested from the growth solution and cleaned by dimethyl ether.

In the dielectric-coating method, the Al$_2$O$_3$ layer was grown on the single crystals by electron-beam evaporation (Denton Infinity 22) with a deposition rate of about 0.4 Å·s$^{-1}$. A clean Si wafer was placed right next to the single crystals as witness wafer for post-calibration of the thickness and the refractive index of the Al$_2$O$_3$ using spectroscopic ellipsometry with a J.A. Woollam M-2000 Ellipsometer. The samples were maintained at ambient temperature during the evaporation process by a water-cooled stage. We note that Al$_2$O$_3$ was chosen due to its high optical transparency over the explored spectral range, as validated by the ellipsometry experiments.

The crystalline thin films of $S$-NPB were fabricated following literature procedures with slight modifications[51]. Specifically, $S$-NPB single crystals were dissolved in DMF at a concentration of 0.2 M. Amorphous films were firstly fabricated by spin-coating the $S$-NPB:DMF solution on precleaned glass substrates at a speed of 3000 rpm for 45 s in an ambient environment. The amorphous films were then annealed at 130 °C for 15 min to induce crystallization.

### Synchrotron X-ray diffraction experiments

Synchrotron X-ray diffraction experiments were performed on the ID4B (QM2) beamline at the Cornell high energy synchrotron source (CHESS). The incident X-ray energy was 31 keV ($\lambda$ = 0.399 Å), which was selected using a double-bounce diamond monochromator. An area detector array (Pilatus 6M) was used to collect the scattering intensities in a reflection geometry. The sample was rotated with three tilted 360° rotations, sliced into 0.1° frames. Geometric parameters of the Pilatus6M detector such as detector distance, tilting, rotation, and direct beam position were extracted using standard CeO$_2$ powder, and data were processed and analyzed using in-house software and visualization performed with the NEXPY software package.

### CD reflectance experiments

A deuterium-halogen lamp (AvaLight-DH-S-BAL, Avantes) was used as the light source. Using achromatic lenses, the lamp output was collimated and focused onto the input slit of a spectrograph (microHR, Horiba), which is equipped with a ruled reflective diffraction grating (1200 grooves·mm$^{-1}$ and 400-nm blaze wavelength; GR25-1204 from Thorlabs). The light from the output slit of the spectrograph passed, in order, a broadband wire-grid linear polarizer (WP25M-UB, Thorlabs) oriented 45° from the horizontal axis, a photoelastic modulator (PEM; I/FS50 and PEM100, Hinds Instruments), a silver half-mirror, before reaching the sample via a broadband reflective objective (10X, NA = 0.22, Edmund Optics).

The light reflected off the sample was focused onto a photo-multiplier tube (H10722-01, Hamamatsu), whose voltage output was sent to a lock-in amplifier (SR860, Stanford Research Systems). The PEM was set to generate one-quarter-wave ($\pi/2$) retardation. The 1$f$ TTL output of the PEM controller provided the reference-in signal for measuring $V_{1f}$ (*i.e.*, the 1$f$ voltage) by the lock-in amplifier. A mechanical chopper (C-995, Terahertz) operating at 140 Hz was placed before the input slit of the spectrograph; the TTL output of the chopper controller provided the reference-in signal for measuring $V_{DC}$ (*i.e.*, the DC voltage) by the same lock-in amplifier. The schematic of the experimental setup is depicted in Supplementary Fig. 2. The spot size of the setup is several hundred micrometers. The spectral resolution of the setup was measured to be less than 1 nm (Supplementary

Fig. 15), which was enabled by controlling the input and output slits of the spectrograph[57].

## Ab initio many-body calculations

We utilized the open-source Quantum Espresso software package with the generalized gradient approximation (GGA) exchange correlation functional in density functional theory (DFT) to compute the ground-state starting point for our GW-BSE calculations[58]. The calculations used norm-conserving, fully-relativistic pseudopotentials from Pseudo Dojo to account for the large spin-orbit coupling effect of Pb[59]. In our structural relaxation process, we applied semiempirical DFT-D3 dispersion corrections to account for the van der Waals interactions[60]. The DFT self-consistent and non-self-consistent calculations employed a 5×5×2 Monkhorst-Pack k-grid and a 60 Ry kinetic energy cutoff. We determined the quasiparticle bandgap through a one-shot GW ($G_0W_0$) calculation performed using the open-source BerkeleyGW package[33]. To expedite self-energy convergence, we applied a static remainder correction for the contributions of higher bands[61]. A 5×5×2 k grid, 10 Ry G-vector energy cutoff and 1650 empty bands were used to calculate the dielectric matrix and self-energy at the GW level. To account for the excitonic effects in the permittivity spectrum, we included electron-hole interactions by solving the Bethe Salpeter equation (BSE) with BerkeleyGW. We used eight conduction bands, eight valence bands and a 24×24×2 k-grid. Convergence with respect to the number of bands in the sum over empty, screened energy cutoff for the polarizability and the number of bands included in the BSE Hamiltonian is shown in Supplementary Fig. 8. The spectrum of BSE eigen-energies colored according to the oscillator strength of each state under different polarizations of light is shown in Supplementary Fig. 7.

We then computed $\Delta\varepsilon''(\omega)$ and $\Delta\varepsilon'(\omega)$ with an in-house code, which includes the magnetic field of the incident light in the perturbation Hamiltonian through the addition of a coupling term $\frac{e}{2m_e}\hat{\boldsymbol{L}} \cdot \boldsymbol{B}$. We used a sum-over-states method to compute the angular momentum ($\hat{\boldsymbol{L}}$) matrix from momentum matrix elements. The details of the calculations are provided in the Supplementary Materials.

## Data availability

Data sets generated during the current study are available from the corresponding author on request.

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

## Acknowledgements

This material is based upon work supported by the National Science Foundation (NSF) under Grant No. CHE- 2305138 and DMR-2313648. Shunran L. and. P.G. acknowledges the partial support from the Air Force Office of Scientific Research (Grant No. FA9550-22-1-0209). This work was partially supported by the donors of ACS Petroleum Research Fund under Doctoral New Investigator Grant 65716-DNI6. P.G. served as Principal Investigator on ACS PRF 65716-DNI6 that provided support for C.A.K.. The theoretical and computational work was primarily supported by the U.S. Department of Energy (DOE), Office of Science, Basic Energy Sciences, under Early Career Award No. DE-SC0021965. This research used the Materials Synthesis and Characterization facility of the Center for Functional Nanomaterials (CFN), which is a U.S. Department of Energy Office of Science User Facility, at Brookhaven National Laboratory under Contract No. DE-SC0012704. Development of the BerkeleyGW code was supported by Center for Computational Study of Excited-State Phenomena in Energy Materials (C2SEPEM) at the Lawrence Berkeley National Laboratory, funded by the DOE Office of Science, Basic Energy Sciences, Materials Sciences and Engineering Division, under Contract DE-AC02-05CH11231, and development of methods used to speed up calculations of the polarizability was supported by the NSF Condensed Matter and Materials Theory (CMMT) program under Grant DMR-2114081. Research conducted at the Center for High-Energy X-ray Science (CHEXS) is supported by the NSF (BIO, ENG and MPS Directorates) under award DMR-1829070. The calculations used resources of the National Energy Research Scientific Computing (NERSC), a DOE Office of Science User Facility operated under Contract DE-AC02-05CH11231; the Texas Advanced Computing Center (TACC) at The University of Texas at Austin; and the INCITE program at the Oak Ridge Leadership Computing Facility, which is a DOE Office of Science User Facility supported under Contract DE-AC05-00OR22725. X.X. and D.Y.Q. thank M.R. Filip and S. Refaely-Abramson for helpful discussions.

## Author contributions

P.G. conceived and supervised the project. Shunran L. synthesized the materials and performed the measurements. X.X. and D.Y.Q. developed the theoretical methods and performed the calculations. C.A.K. set up the helium cryostat for cryogenic optical measurements. C.Z. and M.L. helped with the dielectric coating. Y.L., J.A.B., D.C., Sunhao L. and L.Q. contributed to the sample fabrication. Shunran L. and S.S. performed the X-ray diffraction experiments. P.G., D.Y.Q., X.X., and Shunran L. wrote the manuscript.

## Competing interests

The authors declare no competing interests.
