## [Peer Review File · Nature Communications]

Large exchange-driven intrinsic circular dichroism of a chiral 2D hybrid perovskiteREVIEWER COMMENTS

Reviewer #1 (Remarks to the Author):

The manuscript describes developing a reflection-based approach to characterize the intrinsic, circular polarization-dependent complex refractive index for a prototypical 2D chiral lead-bromide perovskite. An important point is that the previous thin film-based CD measurements largely underestimate the intrinsic chiroptical behavior, likely due to domain variations, poor crystallinity, and defects. Comparison with ab initio theory revealed that the observed band-edge CD originated from excitons residing primarily in the inorganic sublattices and is driven by a purely quantum mechanical electron-hole exchange interaction between degenerate Rashba-Dresselhaus split bands. This work will be of significance to several scientific communities as the investigated interplay of polarization of light, spin of charge carriers, and structural chirality of materials enables exciting properties such as controlled emission and/or detection of circularly polarized light as well as spin-polarized charge transport originating from the chiral-induced spin selectivity effect.

The methodologies are sound, I can not discern any flaws in the data analysis, interpretations, or conclusions. The results support the claims and conclusions, and the details provided would allow the work to be reproduced.

Reviewer #2 (Remarks to the Author):

In this manuscript, the authors characterized the intrinsic circular polarization-dependent complex refractive index of a 2D chiral lead-bromide perovskite by using a reflection-based approach. The authors claimed that by using this method, the true intrinsic circular dichroism can be extracted, which is much larger than that extracted from circular dichroism spectroscopy. The reflection spectroscopy is a very common route to extract the complex refractive index of a semiconductor and the conclusions the authors drawn can not fully supported by the experimental evidences. Therefore, I cannot recommend its publication at the current form due to the lack of the novelty.

1 The reflection spectroscopy is a very common route to extract the complex refractive index of a semiconductor. To extract the circular polarization-dependent complex refractive index of a 2D chiral perovskite, it only need to insert polarizer and waveplate or PEM on the optical path of the measurement system. In this sense, this method is not something new.

2 The authors claimed that CD value of the chiral 2D perovskite is much larger than that extracted by CD spectroscopy since CD spectroscopy requires the measured samples must be thin films, which are due to imperfections and incomplete crystallization. The authors need to explain why imperfections and incomplete crystallization can reduce CD value and provide experimental evidences.

3 It is well known that the reflection spectrum strongly depends on the surface conditions of samples. Can the authors exclude that the large CD value obtained is not from surface of the samples.

4 The authors claimed that 'the CD is an excitonic effect driven by an electron-hole exchange interaction between RD-split states at the band edge'. Under such case, why can we still obtain CD sign far above the band edge?

5 The authors need to explain why they need to use dielectric coating sample as a control sample?

6 In SI, the authors provide the transmittance spectra of amorphous and crystalline samples (Figure S14). How did the authors know that the sample is amorphous? Why the spectra is

so different between them? For the film sample, can the author also provide CD spectrum measured by CD spectroscopy?

7 Can the authors provide the same measurement for ac plane and also for R-NPB and rac-NPB sample for comparisons?

8 The authors need to calibrate their measurement system by using a standard sample to assure their measurement system is reliable.

9 Can the authors evaluate the influence from the fluctuation of the light source and the measurement system when using the waveplate instead of the PEM?

10 In Figure 3a and b, the experimental results deviate from the calculated ones. Can the authors evaluate the origins of such deviation?

11 In Page 7, Figure 3e-f should be Figure 2e-f.

Reviewer #3 (Remarks to the Author):

Please see attached Referee report file.

Referee Report: Nature Communications manuscript NCOMMS-23-39273-T

Title: Large exchange-driven intrinsic circular dichroism of a chiral 2D hybrid perovskite

Authors:

Shunran Li, Xian Xu, Conrad A. Kocoj, Chenyu Zhou, Yanyan Li, Du Chen, Joseph A. Bennett, Sunhao Liu, Lina Quan, Mingzhao Liu, Diana Y. Qiu, Peijun Guo

Executive Summary and Recommendation:

This manuscript presents a combination of experiment and theory pertaining to the intrinsic excitonic circular dichroism (CD) in the chiral 2D hybrid organic-inorganic perovskite compounds, (*S*-NEA)₂PbBr₄ and (*R*-NEA)₂PbBr₄, which they abbreviate as R/S NPB. The Authors claim excitonic CD for single crystal chiral NPB with dissymmetry of order $g_{cd} \sim 0.03$, about 100x what has previously been reported in thin films (very surprising).

The Authors claim that this can be understood as a consequence of Rashba Dresselhaus spin splitting in conjunction with electron-hole exchange creating the CD; and the absence of film defects and imperfections is the reason why their inferred CD in single crystals is so much larger than what has previously been measured in thin films.

Major issues exist that must be addressed before I could recommend publication of this work:

- On the experimental side the Authors' report of CD dissymmetry ~ 100 x larger in single crystal chiral NPB than what has been reported before on thin films begs for a solid explanation and needs to be checked. Some checks were done by the Authors on thin film samples, but a full control study consisting of a work-up of reflectivity data measured on thin film samples, following the same methodology as was used in the analysis of the single crystal samples, was not done as far as I could tell, and should be done.
- On the theoretical side,
 - The Authors do not provide a lot of detail on their calculation. What material they did provide, in the SI, seems to have some issues (what appear to be incorrect expressions for the MD transition matrix elements etc.).
 - They also list what they describe as the necessary conditions to observe CD in such systems:
 - “1) a breaking of the inversion symmetry; 2) strong spin-orbit coupling; 3) the first few conduction bands and valence bands are isolated RD-split bands; 4) a strongly bound and isolated 1st exciton peak; 5) a finite exchange energy among the relevant exciton states.
 - Missing from this list is anything that requires chirality. *All* of the “necessary conditions” listed by the Authors can occur in non-chiral systems.
 - Since chirality is required for CD, something important is missing. The explanation claimed by the Authors for their result of giant CD cannot be correct.

I recommend that the manuscript not be accepted without revisions that address these points. Detailed summary and comments follow.

Summary of the Manuscript:

This manuscript presents a combination of experiment and theory pertaining to the intrinsic excitonic circular dichroism (CD) in the chiral 2D hybrid organic-inorganic perovskite compounds, $(S\text{-NEA})_2\text{PbBr}_4$ and $(R\text{-NEA})_2\text{PbBr}_4$, which they abbreviate as R/S NPB. The Authors claim excitonic CD for single crystal chiral NPB with dissymmetry of order $g_{\text{cd}} \sim 0.03$, about 100x what has previously been reported in thin films. The Authors claim that this can be understood as a consequence of Rashba Dresselhaus spin splitting in conjunction with electron-hole exchange; and the absence of film defects and imperfections.

The experiment consists of reflection-based measurements on large single crystals of R/S NPB which they use to extract the circular-polarization dependent complex refractive index (RI) as a function of the wavelength across the exciton band. Having determined the complex RI functions, the authors then calculate model absorbance and CD (Fig. 4). Remarkably, the Authors find CD values, expressed in ellipticity units, of the order of 1 degree (1000 mDeg) for samples with computed absorbance of the order OD 1 at the exciton line. These numbers correspond to dissymmetry $g_{\text{cd}} \sim 0.03$, which is 100 times larger than what has been reported in the literature for thin film samples of R/S NPB - see Ref. 24, Jana *et al.*, *Nat Comm* **11**, 4699 (2020)). The authors explain this 100x discrepancy as being due to “defects, grain textures, and suboptimal crystallization” in thin films.

The theoretical side of the manuscript comprises, first, a calculation of the DFT band structure with G_0W_0 bandgap starting from the experimental single crystal structure data from Jana *et al.* Ref. 24. The band-structure is given (Fig 3d) along the in-plane line $(-X)\text{-(}\Gamma\text{)}\text{-(}X\text{)}$, which is perpendicular to the in-plane 2_1 screw axis (not stated by the Authors but known from Ref. 24). The band structure shown exhibits Rashba/Dresselhaus (“R/D”) spin splitting along the $\Gamma\text{-}X$ direction with significant spin polarization in the out-of-plane direction, as was previously shown in Jana *et al.* Ref.24.

Second, the Authors report a calculation of the exciton states in chiral NPB using GW-BSE implemented using BerkeleyGW. The exciton states thus found are used to then compute the complex susceptibility function using Fermi’s golden rule, notionally accounting for electric dipole and magnetic dipole terms. Quite remarkable agreement is found between the theory and the experiment (Fig. 3). All details are deferred to the SI.

The explanation of the extremely large CD dissymmetry is given as follows: The independent particle transitions occurring at the Rashba minima at $+K_r$ and $-K_r$ along the line $(-X)\text{-(}\Gamma\text{)}\text{-(}X\text{)}$ have opposite circular polarizations. The Authors claim that electron-hole exchange breaks the degeneracy of these transitions, writing on page 9, “Such exchange interaction splits the degenerate RD exciton states by ~ 17 meV, giving rise to a pronounced derivative-like line-shape in the $\Delta\epsilon''$ spectrum, reminiscent of the Cotton effect”. The Authors claim that this exchange splitting is critical writing that without this, “Due to the degeneracy of these two sets of transitions, contrary to previous hypotheses that the RD splitting alone is responsible for chirality transfer between sublattices [24, 46] we would not expect to observe CD”.

The requirements to observe CD are further elaborated in the SI, where the Authors write on page S18:

“We list several necessary conditions for the formation of the exchange-driven Cotton effect: 1) a breaking of the inversion symmetry; 2) strong spin-orbit coupling; 3) the first few conduction bands and valence bands are isolated RD-split bands; 4) a strongly bound and isolated 1st exciton peak; 5) a finite exchange energy among the relevant exciton states.

Analysis and Comments- experimental

First I will comment in detail on the experimental side of the work.

As noted above, remarkably, the Authors infer from their measurements and analysis of reflectivity data on single crystals that CD values for chiral NPB, expressed in ellipticity units, would be of the order of 1 degree (1000 mDeg) for samples with computed absorbance of the order OD1 at the exciton line. These numbers correspond to dissymmetry $g_{cd} \sim 0.03$, which is 100 times larger than what has been reported in the literature for thin film samples of R/S NPB - see Ref. 24, Jana *et al.*, *Nat Comm* **11**, 4699 (2020)). Specifically, Jana et al reported excitonic CD of ~ 20 -30 mDeg on oriented thin film samples of R/S NPB with absorbance of order OD2 at the exciton corresponding to $g_{cd} \sim 0.0003$; 100 times smaller than what the Authors calculate on the basis of their single crystal RI measurements.

The authors explain this 100x discrepancy as being due to “defects, grain textures, and suboptimal crystallization” in thin films. They support this explanation as follows:

- The Authors show that the single crystal samples have high index XRD peaks that are absent in XRD of thin films (both theirs and Jana’s), from which it can naturally be inferred that the single crystal samples have a higher degree of order and fewer defects.
- The Authors also show measurements of circular polarized reflectivity on thin films of S-NPB of their own making to show the inferiority of the thin film samples. Actually, these measurements (Fig 5) look quite similar to what they show in Fig 4: Similar reflectivity spectra and similar polarization dependence and depth of modulation with waveplate angle.

Due to the last point (similarity between reflectivity data on films versus single crystals) I found the Authors’ explanation of 100x larger dissymmetry relative to prior reports on thin films unconvincing. I see nothing in what was presented that could explain a 100x discrepancy in the g_{cd} values they infer from their reflectivity measurements versus what has been measured in thin films.

Recommendation: It would have been much better to perform a full *control work-up* of reflectivity data measured on thin film samples, following the same methodology as the Authors used in their analysis of the single crystal samples, to see if the results from analysis of reflectivity data on thin films gives estimated g_{cd} in line with prior measurements. This would support their hypothesis. If this was done it was not described that I could find.

Analysis and Comments- Theoretical.

The exciton calculations described by the authors are done in GW-BSE on chiral NPB, which has 118 atoms per unit cell. Such a calculation is a significant undertaking and represents a significant accomplishment if it were done correctly. I haven't seen such BSE calculations in the literature previously for fully realistic hybrid organic-inorganic perovskites ... only simplified model systems using Cs ions as stand-ins for the organic cations (see e.g., Quarti et al., *Adv. Optical Mater.* **2023**, 2202801). Consequently I was surprised at how little detail was provided by the Authors on this calculation. I would have expected to see a report on the exciton binding energy, excitation radius; fine structure levels, etc. However, none of this was discussed.

The only statement given by the Authors about the exciton is the claim that electron hole exchange splits the LCP and RCP exciton states – which correspond to the two in-plane bright exciton levels -- by 17 meV.

This statement in itself is quite surprising.

The effect of electron-hole exchange on the exciton fine structure has been calculated in model 2D layered lead bromides: With single monolayer PbBr₄ octahedra layers the computed bright-dark splitting in GW-BSE is ~ 30 meV (see Fig 2 in *Adv. Optical Mater.* **2023**, 2202801); experiments have shown ~ 28 meV (Dyksik et al., *Sci. Adv.* **7**, eabk0904 (2021)) with minimal splitting of the in-plane polarized exciton fine structure levels.

So it seems unlikely that electron-hole exchange, per se, could be responsible for the claimed 17meV in-plane polarized bright exciton splitting.

Could the source of the splitting be the Rashba/Dresselhaus splitting? As the authors correctly claim, the R/D term that corresponds to the spin splitting and spin texture shown in Fig 3d does not split the transitions from the +/- K_r valleys in an independent particle picture.

However it can *also* be shown that this R/D term does not split the bright in-plane polarized *exciton*, even with electron-hole exchange. This can be shown by analysis of the effect of the spin splitting on the exciton fine structure in 2D layered systems, including electron-hole exchange, within second order perturbation theory...which is almost certainly justified given the expected large exciton binding energy in monolayer thickness PbBr layered perovskites.

So it is not clear, to me, what is going on in this calculation. Few details are provided to enable disentangling.

We can however examine the calculation from the standpoint of what went into it with respect to symmetry. The Authors claim that the necessary conditions to observe CD (from SI page S18):

“We list several necessary conditions for the formation of the exchange-driven Cotton effect: 1) a breaking of the inversion symmetry; 2) strong spin-orbit coupling; 3) the first few conduction bands and valence bands are isolated RD-split bands; 4) a strongly bound and isolated 1st exciton peak; 5) a finite exchange energy among the relevant exciton states.

Missing from this list is anything that requires chirality.

All of the “necessary conditions” listed by the Authors can occur in non-chiral systems.

For example, the Authors emphasize the importance of the RD splitting shown in Fig 3d. This R/D type spin splitting is due to bulk inversion asymmetry which creates linear-in-K spin splitting according to the invariant, (see Jana *et al.*, *Nat Comm* 2021, 12:4982),

$$H_{\text{BIA}} = \alpha_{zx} S_z K_x.$$

Here, S_z and K_x are spin operators and wavevector respectively along the Z and X directions. This invariant gives spin polarization in the out-of-plane (z) direction for wavevector in the in-plane X direction, normal to the 2_1 screw axis, which is what the authors show in Fig 3d.

However, this R/D term exists in non-chiral systems, as does electron-hole exchange obviously.

So there is something critical missing from the Authors' list of "necessary conditions". Again: nothing that the authors list above as critical ingredients requires *chirality*, and yet we know that intrinsic CD requires chirality. Consequently either there is physics going on in the GW-BSE calculation that the Authors have not conveyed, or, there is some other issue.

What details on the calculation that are provided are largely in the SI. Here I found several problems.

1. In SI Note 4 the authors provide a simple analysis based on Fermi's golden rule for the circular polarization dependent permittivity. They arrive at an equation for the CD as follows:

$$\Delta\varepsilon''(\omega) = \frac{16\pi^2 e^2}{\omega^2} \sum_{vck} \text{Im}(X_E^x X_M^{y*} - X_E^y X_M^{x*}) \delta(\omega - \omega_{vck})$$

where X_E^i and X_M^i are components of the electric and magnetic transition dipole matrix elements. This looks to me as though the Authors are saying that $\text{CD} \sim \text{Im}[\vec{X}_E \times \vec{X}_M]$, ie that the CD should go like the *cross product* of the electric and magnetic transition dipoles.

If this expression was actually used in the computations reported in this manuscript, I do not know what they are calculating. Unless I am misunderstanding something in a big way, the given equation is incorrect. It is well known that CD goes as the inner product of the electric and magnetic transition matrix elements, $\text{CD} \sim \text{Im}[\vec{X}_E \cdot \vec{X}_M]$. See Andrews and Tretton, J. Chem. Educ. 2020, 97, 4370–4376, where a similar FGR analysis leading to the expression for the CD response is given and which shows this quite clearly.

2. The computation of the magnetic dipole transition matrix element is known to be challenging. The Authors in Note 4 explain how they calculate it: They discard the spin portion and retain the orbital angular momentum portion. The expression that they give for the magnetic dipole transition matrix element is given on SI page S14:

We can then apply a sum-over-states method to calculate the angular momentum matrix elements from the momentum matrix elements.^{13, 14, 15}

$$L_x(\mathbf{k}; m, n) = \frac{i\hbar}{m_e} \sum_{n'} \left(\frac{\langle \mathbf{k}m | p_y | \mathbf{k}n' \rangle \langle \mathbf{k}n' | p_z | \mathbf{k}m \rangle}{E_{mk} - E_{n'k}} - \frac{\langle \mathbf{k}m | p_z | \mathbf{k}n' \rangle \langle \mathbf{k}n' | p_y | \mathbf{k}m \rangle}{E_{mk} - E_{n'k}} \right).$$

L_y and L_z can be calculated in a similar fashion.

The problem is that this expression gives something proportional to the *orbital magnetic moment of a given state*, it is *NOT a transition matrix element*. The left hand side of the expression involves band indices m,n; the right hand side does not involve the index n. The equation is simply incorrect. All the references given (Refs 13-15 in the SI) pertain to the calculation of the Lande g-factor of carriers, NOT to magnetic dipole transition matrix elements.

So, if the given expression was actually used in the computations reported in this manuscript, I do not know what they are calculating, but it is not recognizable to me as CD.

3. Minor issues:

- a. Here and there in the Authors' mathematical expressions the terms $(\nabla \times \hat{\epsilon})$ appears. For example on Page S13:

$$X_M^{vck} = \frac{1}{2} \langle v\mathbf{k} | \hat{\mathbf{L}} \cdot (\nabla \times \hat{\epsilon}) | c\mathbf{k} \rangle$$

Since $\hat{\epsilon}$ is defined in Note 4 as a unit vector, this term vanishes. Such expressions are missing something and this should be fixed.

- b. Different definitions are used for right and left polarization vectors. For example in Note 4 the definition are given $\hat{\epsilon}_L = (\hat{x} + i\hat{y})/\sqrt{2}$, which is correct according to IUPAC convention. But in Note 3 the opposite definition is given: $\hat{\epsilon}_L = (\hat{x} - i\hat{y})/\sqrt{2}$
This sort of inconsistency should be corrected.

Other minor issues.

On page 9 the Authors write: “.. contrary to previous hypotheses that the RD splitting alone is responsible for chirality transfer between sublattices [24, 46] we would not expect to observe CD”.

I think that the Authors here mean that RD splitting cannot be responsible for CD. Ref 24 made no such claim (although Ref 46 did...and the claim is not correct). Ref 24 is a study on the mechanism of chirality transfer from the chiral organics to the inorganic sublattice -- which results in RD splitting of bands associated with the inorganic sublattice. Ref 24 is agnostic about the mechanism of CD.

Editorial Note: Figure R3a of this Peer Review File have been redacted as indicated to remove third-party material where no permission to publish could be obtained.

Reviewer #1

The manuscript describes developing a reflection-based approach to characterize the intrinsic, circular polarization-dependent complex refractive index for a prototypical 2D chiral lead-bromide perovskite. An important point is that the previous thin film-based CD measurements largely underestimate the intrinsic chiroptical behavior, likely due to domain variations, poor crystallinity, and defects. Comparison with ab initio theory revealed that the observed band-edge CD originated from excitons residing primarily in the inorganic sublattices and is driven by a purely quantum mechanical electron-hole exchange interaction between degenerate Rashba-Dresselhaus split bands. This work will be of significance to several scientific communities as the investigated interplay of polarization of light, spin of charge carriers, and structural chirality of materials enables exciting properties such as controlled emission and/or detection of circularly polarized light as well as spin-polarized charge transport originating from the chiral-induced spin selectivity effect.

The methodologies are sound, I cannot discern any flaws in the data analysis, interpretations, or conclusions. The results support the claims and conclusions, and the details provided would allow the work to be reproduced.

We would like to thank the Reviewer for their positive feedback on our work and its potential impact on a range of chiral perovskite-related phenomena and applications.

Reviewer #2

In this manuscript, the authors characterized the intrinsic circular polarization-dependent complex refractive index of a 2D chiral lead-bromide perovskite by using a reflection-based approach. The authors claimed that by using this method, the true intrinsic circular dichroism can be extracted, which is much larger than that extracted from circular dichroism spectroscopy. The reflection spectroscopy is a very common route to extract the complex refractive index of a semiconductor and the conclusions the authors drawn cannot fully supported by the experimental evidence. Therefore, I cannot recommend its publication at the current form due to the lack of the novelty.

We thank the Reviewer for reviewing our work. In the revision, we carefully addressed all the comments, as discussed below.

1. The reflection spectroscopy is a very common route to extract the complex refractive index of a semiconductor. To extract the circular polarization-dependent complex refractive index of a 2D chiral perovskite, it only needs to insert polarizer and waveplate or PEM on the optical path of the measurement system. In this sense, this method is not something new.

We thank the Reviewer for pointing this out. When discussing reflection spectroscopy-based modeling of complex refractive index (RI), it is important to clarify how the modeling is executed.

It is true that reflection spectroscopy was used to extract RI, with a classic paper being *Phys. Rev. B.* **2014**, *90*, 205422. Spectroscopic ellipsometry is another type of widely used, reflection-based spectroscopy technique to determine the RI. However, it is worth noting that all these approaches rely on using some type of model (e.g., Drude model, Cauchy model, etc.), where the wavelength dependence of permittivity is prescribed by several fitting parameters. Extracting the RI is equivalent to varying those fitting parameters over a large range and searching for a best fit to experimental data.

However, we believe that the extraction of complex RI from reflection spectroscopy using a “model-blind” (i.e., no spectroscopic modeling) approach, as we demonstrated in this work, is quite unique. The only other work (to our knowledge) was that published by our group earlier, in *Phys. Rev. Rev.* **2018**, *121*, 127401, on non-chiral 2D perovskites. Such “model-blind” approach is advantageous because it offers higher accuracy (i.e., there is essentially zero fitting error), although it comes at a cost that two independent reflectance measurements are needed, as opposed to prior model-based approaches where a single reflectance measurement can be sufficient (e.g., *Phys. Rev. B.* **2014**, *90*, 205422).

Then, regarding the determination of circular polarization (CP)-dependent complex RI: inserting a polarizer will likely not work, as the $1f$ signal (proportional to the CD response) is typically very small. Lock-in based detection is necessary to pick up the small signal that is vulnerable to noise and drifts associated with the detector and the light source. As such, inserting a PEM is necessary for these experiments. However, it gets much more subtle to extract the CP-dependent RI after adding the PEM. This arises because there are no pre-established models for describing (or, in a sense, predicting) the variation of RI with a change in CP, making the above “model-based” approach unamenable to extract the CP-dependent RI. It is only with our “model-blind” approach, that we can determine the CP-dependent RI, with which we learned, for the first time, that the imaginary RI varies by adopting a derivative-like line shape, and the variation in the real RI consists of two parts, including a constant shift suggesting circular birefringence, and a Kramers-Kronig contribution coming from the imaginary RI. These forms of variations of RI with CP would not have been known a priori. For these reasons, we believe that our determination of CP-resolved, complex RI is unique and hasn’t been done for any chiral perovskites.

2. The authors claimed that CD value of the chiral 2D perovskite is much larger than that extracted by CD spectroscopy since CD spectroscopy requires the measured samples must be thin films, which are due to imperfections and incomplete crystallization. The authors need to explain why imperfections and incomplete crystallization can reduce CD value and provide experimental evidence.

We thank the Reviewer for this important question. In accordance with our reply to Reviewer 3, we performed additional CD reflectance experiments on annealed thin films using our setup

developed in this work. As expected, we found the films exhibit lower V_{1f}/V_{DC} values (a proxy to the CD value) compared with single crystals; these data are added to the revised version in Fig. 4e, 4f for S-NPB and Fig. S16 for R-NPB.

We hypothesized that there can be three possible reasons for the observed lower CD for annealed films compared to single crystals. **1)** Lower crystallinity of the annealed films. To further support this point, we performed reciprocal space mapping x-ray diffraction experiments at a synchrotron facility. These new results, which now appear in Fig. S17 of the revised manuscript, revealed significant difference in the crystallinity between single crystals and thin films. The non-crystallized region in the films does not contribute to the CD near the exciton resonance, hence reducing the CD value of the films. **2)** Our *ab initio* theoretical results suggest that the selection rules imposed by the spin texture and RD splitting underpin the large CD around the exciton resonance. In other words, the Cotton effect is very sensitive to the energy splitting of the two exciton states, which is in turn very sensitive to the crystal structure. In conjunction with point 1), we anticipate that lowered crystallinity, high-density grain boundaries (*e.g.*, those between domains of randomly distributed in-plane orientations), and defects present in the films can significantly relax the selection rules, thereby reducing the CD effects. **3)** A close inspection of the synchrotron x-ray diffraction data (Supplementary Fig. 17) further reveals that the annealed film has a slightly different lattice constant (~0.4% contraction) along the *c* direction (*i.e.*, perpendicular to the 2D perovskite layers) from that of single crystals. This may lead to less amount of octahedral tilting distortion, and with it weaker chiroptical effects (*Nat. Commun.* **2021**, *12*, 4982). Since the reported crystal structure is solved with single-crystal diffraction data, the true octahedral tilting distortion in annealed polycrystalline films may be lower than single crystals. In the revised manuscript, we expanded our manuscript by adding the points discussed above, which now appear before the conclusion paragraph. This section has been largely modified and can be found in the main text file with tracked changes (not repeated here in observation of the length of this response letter).

We'd like to note that our present manuscript provides a first experimental demonstration of intrinsic CD extraction for single crystals, illustrates the difference between crystals and films, and offers a full theoretical account of the intrinsic CD response of the chosen chiral 2D perovskite. However, it is likely that to fully and quantitatively understand the differences between thin films and single crystals, more future work is warranted. These efforts may include nanoscale x-ray diffraction or transmission electron microscopy to reveal the local crystalline order and octahedral tilting distortions of thin films in comparison to single crystals, investigation of more chiral perovskite compounds with varying degrees of crystallinities exhibited by thin films, and scanning tunneling microscopy approaches to reveal spatial heterogeneity in the response to spin-polarized charges, to name a few.

3. It is well known that the reflection spectrum strongly depends on the surface conditions of samples. Can the authors exclude that the large CD value obtained is not from surface of the samples.

Our optical micrograph and scanning electron microscopy images (Fig. S1) demonstrate that the crystals are extremely smooth and uniform. Apart from intentionally included small features on the surface to demonstrate the pictures were taken at focus (Fig. S1b), the crystal exhibit extremely high single-crystalline quality. When light impinges on the surface on these high-quality single crystals, we are indeed accessing the intrinsic properties of the material, not affected by any roughness or scatterers that may affect the results. We modified the main text to read: “reveal the high quality of centimeter-sized single crystals exhibiting optically smooth *a-b* facets parallel to the lead-bromide layers, allowing us to access the intrinsic optical properties of these materials.”

4. The authors claimed that ‘the CD is an excitonic effect driven by an electron-hole exchange interaction between RD-split states at the band edge’. Under such case, why can we still obtain CD sign far above the band edge?

We apologize for the lack of clarity here. We meant specifically that the observed Cotton effect in the CD at the band edge is driven by the electron-hole exchange interaction, which significantly enhances CD at the band edge. We have modified the main text to read: “Theoretical calculations show that the Cotton effect seen in the CD spectrum and enhanced CD at the band edge are excitonic effects driven by an electron-hole exchange interaction between RD-split states at the band edge ...”.

5. The authors need to explain why they need to use dielectric coating sample as a control sample?

We’d like to clarify that the dielectric coating sample is not a control sample in a traditional sense. Rather, it was a sample we fabricated and utilized for our determination of the complex refractive index (RI). Briefly, at each wavelength, the perovskite’s complex RI has a real part (n) and an imaginary part (k). To determine these two unknowns of n and k , we need two input parameters, which in our approach are the reflectivity values of the perovskite before the coating (R_0) and after the coating (R_1). Without the dielectric-coated perovskite sample, we would not have enough information to determine both n and k , unless we adopt permittivity models. In fact, this point is related to our response to the first question, where we stated that our approach is “model-blind”, simply because there are two input parameters, sufficient to determine the two unknowns (n and k). We have modified our text to read “As a “model-blind” approach that does not require parametric fitting, the method involves two separate reflectivity measurements...”

6. In SI, the authors provide the transmittance spectra of amorphous and crystalline samples (Figure S14). How did the authors know that the sample is amorphous? Why the spectra are so different between them? For the film sample, can the author also provide CD spectrum measured by CD spectroscopy?

We apologize for not including an x-ray diffraction data for the amorphous sample, as we found it featureless. In the revised version, we added x-ray diffraction data for the amorphous sample to Fig. S15. The amorphous-crystalline transition for the compound we studied in this manuscript is described in detail in the paper *Adv. Mater.* **2021**, *33*, 2005868, published by David Mitzi’s group

(ref. 19 in our manuscript), which also discusses the difference in transmittance spectra between the two phases. Our results on the amorphous-crystalline phases are consistent with their report.

For the revision, we performed CD reflectance and traditional CD transmittance experiments on film samples, now appearing in Fig. 4 and Fig. S16.

7. Can the authors provide the same measurement for *ac* plane and also for R-NPB and rac-NPB sample for comparisons?

The measurement setup we developed has a spot size on the order of 300~500 μm (due to the use of the spectrograph and the PEM), which is much larger than the area of the *ac* plane of these layered materials (see Fig. S1; the thickness is typically tens of μm). The *ac* plane is also much less smooth than the *ab* plane, which is quite common for 2D layered materials. For these reasons, we cannot perform measurements on the *ac* plane at the moment.

However, we note that for 2D perovskites the exciton effects are much stronger with light incidence on the *ab* plane than on the *ac* plane (*Phys. Rev. Lett.* **2018**, *121*, 127401; *Phys. Rev. B* **1990**, *42*, 11099, Fig. 4a), and they typically adopt an out-of-plane orientation with respect to the substrates. As such, our primary focus of this work is on the response of the *ab* plane, addressing the exchange interaction-induced CD effect. We hope to perform future studies on other 2D or even 1D chiral perovskites that may exhibit larger facets with different orientations.

We indeed performed experiments on R-NPB, although we showed most of the results pertaining to R-NPB in the Supplementary Information, as these results are opposing, but similar to the results on S-NBP. We also examined rac-NPB but did not observe any CD response. To clarify this point, we added a sentence, which reads as “Note that we did not observe noticeable lock-in voltage at the 1f frequency for racemic-NPB crystals, suggesting its lack of CD response”.

8. The authors need to calibrate their measurement system by using a standard sample to assure their measurement system is reliable.

To our knowledge, there is no well-accepted standard sample in this type of measurements (unlike, *e.g.*, photoluminescence quantum yield measurement).

In the biochemistry field, CD experiments are mainly performed on biomolecules in solvents, but the volume fraction of the biomolecules relative to the solvent is extremely small. As a result, the optical response of biomolecules in solvents is mainly dominated by absorption, not scattering, and reflectance measurement is not suitable (nor is the extraction of refractive index meaningful).

For solid-materials research, as we show in the paper the CD response is very material-processing dependent. As such, we believe the best “standard sample” available to test our setup is to use commercially available optics to make a sample with a definitive response to circularly polarized light. To verify our CD reflection spectroscopy, we leveraged a quarter-wave plate for converting circularly polarized (CP) reflected light to linear polarized light. Specifically, as depicted below, we used a combination of silver mirrors, quarter-wave plates, and linear polarizers

to simulate an ideal CP reflector. By properly setting the angle of the optical axis of the linear polarizer and the quarter waveplate This reflector can have a 100% reflectance for LCP (RCP) but a 0% reflectance for RCP (LCP).

According to our derivation, the square of the electric field amplitude received by the detector is:

$$|E^{(3)}|^2 = \frac{E^2}{2} \{(R_R + R_L) + [1.1334 \cdot \sin(2\pi ft) + 0.1378 \cdot \sin(6\pi ft) + 0.0044 \cdot \sin(10\pi ft)](R_R - R_L)\}$$

In our benchmark measurements, we set the linear polarizer and quarter-wave plate such that $R_R = 1$ and $R_L = 0$. So the equation above can be rewritten as following:

$$|E^{(3)}|^2 = \frac{E^2}{2} \{1 + [1.1334 \cdot \sin(2\pi ft) + 0.1378 \cdot \sin(6\pi ft) + 0.0044 \cdot \sin(10\pi ft)]\}$$

The ideal ratio of the V_{pp} (peak-to-peak voltage) of the $1f$ and the DC component resolved by lock-in amplifier should be 1.1334:1. The experimentally measured ratio of the DC and the $1f$ component was $37.33:33.05 = 1.1295:1$, which is very close to the theoretical ratio.

9. Can the authors evaluate the influence from the fluctuation of the light source and the measurement system when using the waveplate instead of the PEM?

Per our response to the 3rd Reviewer, when we used a waveplate in replacement of the PEM, we noticed that there was strong spot-to-spot variation in the measurements on films. This spot-to-spot variation arises owing to **1**) the intrinsic spatial inhomogeneity of the films, and **2**) the very small spot size of the waveplate-based measurements (a couple of μm , since we could use a high NA objective), compared to the hundreds-of- μm spot size in PEM-based measurements. In the original version of the manuscript, we indeed pointed out the spatial inhomogeneity of the films, and we only showed results for those spots with the highest CD response after many experiments.

After repeating these experiments during the revision, we found that those spots with the highest CD response were associated with some stripe-like domains formed on the films. These domains were rare, but not absent, across the films. The bulk part of the film, however, shows much lower CD values, which we now present in Fig. 4e and Fig. 4f in the revised manuscript. We choose to leave out the previous data measured with the waveplate on the minor, stripe-like

domains, which are not truly representative of the films. Note that the larger spot size (hundreds of μm) in our PEM-based experiments on both films and single crystals dictates that the results are always acquired from spatial averages.

10. In Figure 3a and b, the experimental results deviate from the calculated ones. Can the authors evaluate the origins of such deviation?

Overall, we would argue that the experimental and calculated results presented in Fig. 3 agree quite well with the usual margin of error for calculations of this nature. There are small deviations in the width of the spectral features in Fig. 3b, which come from two main sources. Firstly, in our *ab initio* calculations, we only obtain the exciton wavefunctions and excitation energies. We cannot directly calculate the lifetime, which gives rise to the broadening. Hence, a constant broadening of 46 meV was included as an experiment-informed empirical parameter. Secondly, the magnitude of the Cotton effect is very sensitive to the energy splitting of the two exciton states, which is in turn very sensitive to the crystal structure. Relaxation of the crystal structure using the PBE functional gives rise to a 3.4% change in the lattice constant compared to the experimental structure, which also affects the splitting of the exciton states.

11. In Page 7, Figure 3e-f should be Figure 2e-f.

We have rectified this error in the revised manuscript.

Reviewer #3

This manuscript presents a combination of experiment and theory pertaining to the intrinsic excitonic circular dichroism (CD) in the chiral 2D hybrid organic-inorganic perovskite compounds, (*S*-NEA)₂PbBr₄ and (*R*-NEA)₂PbBr₄, which they abbreviate as *R/S* NPB. The authors claim excitonic CD for single crystal chiral NPB with dissymmetry of order $g_{\text{cd}} \sim 0.03$, about 100x what has previously been reported in thin films (very surprising).

The Authors claim that this can be understood as a consequence of Rashba-Dresselhaus spin splitting in conjunction with electron-hole exchange creating the CD; and the absence of film defects and imperfections is the reason why their inferred CD in single crystals is so much larger than what has previously been measured in thin films. Major issues exist that must be addressed before I could recommend publication of this work.

I recommend that the manuscript not be accepted without revisions that address these points. Detailed summary and comments follow.

We thank the Reviewer for their detailed and thoughtful comments, which have helped us improve the manuscript. In response to the constructive feedback, we have fixed a number of typographical errors and greatly expanded the theoretical discussion and analysis to enhance the clarity.

Firstly, we would like to emphasize that the exchange interaction we observe between the Rashba-split states is responsible for the Cotton effect feature in the CD at the band edge, not for the existence of the CD in general (as seen in Fig. S9e), which requires that the crystal belong to a chiral space group. We modified the manuscript to emphasize this point.

Secondly, the Reviewer cites previous work discussing the exchange interaction in perovskites. We would like to point out that our results are indeed completely consistent with previous results, which also observed a splitting of the bright in-plane exciton states once octahedral tiling is taken into account (as it is in our calculations). Our new insight here is the relationship between this splitting, which we find is not simply due to the crystal field but also requires the electron-hole exchange interaction.

Finally, we also greatly appreciate the Reviewer's point that GW-BSE calculations on the full layered perovskite structure are rare and a significant accomplishment in themselves. We note that our group has published such calculations in the past (doi.org/10.1021/acs.nanolett.3c00082, doi.org/10.1021/acs.nanolett.2c01306), though this is the first time we have done so for chiral perovskites. We reply to the Reviewer's individual comments in greater detail below.

(a) On the experimental side the Authors' report of CD dissymmetry ~ 100 x larger in single crystal chiral NPB than what has been reported before on thin films begs for a solid explanation and needs to be checked. Some checks were done by the Authors on thin film samples, but a full control study consisting of a work-up of reflectivity data measured on thin film samples, following the same methodology as was used in the analysis of the single crystal samples, was not done as far as I could tell, and should be done.

We thank the Reviewer for this insightful and important comment. For the revision, we conducted additional experiments and analysis. We reply to this comment in greater detail below.

Experimental aspect

Comment

As noted above, remarkably, the Authors infer from their measurements and analysis of reflectivity data on single crystals that CD values for chiral NPB, expressed in ellipticity units, would be of the order of 1 degree (1000 mDeg) for samples with computed absorbance of the order OD1 at the exciton line. These numbers correspond to dissymmetry $g_{cd} \sim 0.03$, which is 100 times larger than what has been reported in the literature for thin film samples of R/S NPB – see Ref. 24, Jana *et al.*, Nat. Commun. 11, 4699 (2020). Specially, Jana *et al.* reported excitonic CD of ~ 20 -30 mDeg on oriented thin film samples or R/S NPB with absorbance of order OD2 at the exciton corresponding to $g_{cd} \sim 0.0003$; 100 times smaller than what the Authors calculate on the basis of their single crystal RI measurements.

The authors explain this 100X discrepancy as being due to “defects, grain textures, and suboptimal crystallization” in thin films. They support this explanation as follows:

a) The Authors show that the single crystal samples have high index XRD peaks that are absent in XRD of thin films (both theirs and Jana's), from which it can naturally be inferred that the single crystal samples have a higher degree of order and fewer defects. b) The Authors also show measurements of circular polarized reflectivity on thin films of S-NPB of their own making to show the inferiority of the thin film samples. Actually, these measurements (Fig. 5) look quite similar to what they show in Fig 4: Similar reflectivity spectra and similar polarization dependence and depth of modulation with waveplate angle.

Due to the last point (similarity between reflectivity data on films versus single crystals) I found the Authors' explanation of 100X larger dissymmetry relative to prior reports on thin films unconvincing. I see nothing in what was presented that could explain a 100 discrepancy in the g_{cd} values they infer from their reflectivity measurements versus what has been measured in thin films.

Recommendation: It would have been much better to perform a full control work-up of reflectivity data measured on thin film samples, following the same methodology as the Authors used in their analysis of the single crystal samples, to see if the results from analysis of reflectivity data on thin films gives estimated g_{cd} in line with prior measurements. This would support their hypothesis. If this was done it was not described that I could find.

Point a) is addressed in our response to the 2nd comment of Reviewer 2, which summarizes our hypotheses for the lower CD values of annealed films in comparison to single crystals. Briefly recapitulating, we think there may be three possible effects for the annealed films: 1) lower amount of crystalline domains for producing the CD response (backed up by old and newly obtained XRD data); 2) as an indirect consequence, the poor crystallinity and random in-plane-oriented domains & interfaces relax the selection rules that underpins the CD effect observed for single crystals (backed up by XRD data and theory); 3) a subtle difference in the crystal structure and possibly less degree of octahedral tilting distortion compared to single crystals (partially backed up by XRD data; more future work warranted). We believe these effects may be combined to contribute to the large difference in dissymmetry factor between thin films and single crystals. Additional discussion has been added to the revised manuscript (see our response to Reviewer 2, comment 2).

Point b) is largely addressed in our response to the 9th comment of Reviewer 2. Here we present a picture for the rarely occurring stripe-like domains that we saw on the annealed films, where we took the CD reflectivity data on in our original version of the manuscript, referred to by the Reviewer in this comment (and we fully agree that the data does not explain a 100 times discrepancy from single crystals in terms of the g_{cd} values). As we answered above to Reviewer 2, we choose to leave out the data on these rarely occurring, inhomogeneous domains (which sometimes have a large CD response), and show results on the more representative, "featureless" regions on the films.

The featureless film regions showed lower CD response. In the revised manuscript, the newly acquired results on the featureless film appeared in Fig. 4e-4f (for S-NPB) and Fig. S16 (for R-NPB). The CD spectra we measured on our films are somewhat higher than Jana's work. Nevertheless, the film's V_{1f}/V_{DC} (Fig. 4f) measured with our reflectance setup is about 10 times lower than single crystal data (Fig. 2a), which is consistent with the expected order of magnitude larger CD value for single crystals than films. We also note that in a recent publication (*Sci. Adv.* **2020**, *6*, eabd3274), large CD values (up to 3 deg in ellipticity) have been observed, suggesting (at least) the possibility of large CD responses from chiral perovskites.

Theoretical aspect

Comment 1

The exciton calculations described by the authors are done in GW-BSE on chiral NPB, which has 118 atoms per unit cell. Such a calculation is a significant undertaking and represents a significant accomplishment if it were done correctly. I haven't seen such BSE calculations in the literature previously for fully realistic hybrid organic-inorganic perovskites ... only simplified model systems using Cs ions as stand-ins for the organic cations (see e.g., Quarti et al., *Adv. Optical Mater.* **2023**, *2202801*). Consequently, I was surprised at how little detail was provided by the Authors on this calculation. I would have expected to see a report on the exciton binding energy, excitation radius, fine structure levels, etc. However, none of this was discussed. The only statement given by the Authors about the exciton is the claim that electron-hole exchange splits the LCP and RCP exciton states – which correspond to the two in-plane bright exciton levels – by 17 meV. This statement in itself is quite surprising.

We thank the Reviewer for noting that the GW-BSE calculation on the full structure (as opposed to a simplified model with Cs atoms replacing the cations) is a significant accomplishment. We note that we have published a number of similar GW-BSE results analyzing the effect of replacing the molecular cation with Cs cations in layered perovskites of comparable size in the past [doi.org/10.1021/acs.nanolett.3c00082, doi.org/10.1021/acs.nanolett.2c01306], so such a calculation is far from unusual. Because the focus of the paper was on the new experimental measurements of CD, we initially did not include a large amount of detail on the theoretical results, but we greatly appreciate the Reviewer's point that presenting more analysis might be of interest to the readers.

Below, in Fig. R1a-d, we show convergence of the GW and BSE calculations with respect to the screened cutoff, the number of empty states, and the number of valence and conduction bands used to solve the BSE Hamiltonian. In Fig. R1e-f, we show more detailed analysis of the exciton eigenstates (Fig. R1e depicts the first bright exciton in the real space with a 1.42-nm radius). The binding energy associated with the first bright exciton is quantified at 0.46 eV. Fig. R1f shows that these exciton states originate from the transitions between the lowest conduction band and the highest valence band, which are the RD-split states.

Figure R1 | Computational details. **a** Convergence of the G_0W_0 energy gap at the Γ point with respect to the total number of bands included in the sum over empty states in the calculation of the GW self-energy and the screened energy cutoff for the dielectric matrix. **b** Convergence of the G_0W_0 energy gap at the Γ point with respect to the total number of bands with and without the static remainder correction.⁹ **c** and **d** Convergence of the imaginary part of the macroscopic dielectric function with the number of valence bands (n_v) in **c** and conduction bands (n_c) in **d**. **e** First bright exciton in the real space with a 1.42-nm radius and 0.46-eV binding energy. **f** Contribution of each band to each exciton state. The size of each dot corresponds to $f \times \sum_{nk} |A_{vck}^S|$, where A_{vck}^S is the electron-hole amplitude of the exciton state $|S\rangle = \sum_{vck} A_{vck}^S |vck\rangle$, \mathbf{k} is a k -point, and $v(c)$ is the index of the hole (electron) state, as counted from the Fermi energy, contributing to the exciton state. For the conduction (valence) state contributions, n in the sum runs over valence (conduction) states. f is the oscillator strength of each exciton state under excitation by RCP.

In Fig. R2, we show the eigenvalues of the first four exciton states with the spectrum colored according to the oscillator strength under excitation by linear and circularly polarized light. The top row shows the full spectrum from the BSE calculation, while the bottom row shows the spectrum in the absence of an electron-hole exchange interaction. The two low-energy bright excitons in the circularly polarized spectra, which exhibit a splitting of 17 meV, correspond to the excitons discussed in the main text. Fig. R2a shows that these two bright exciton states arise from two excitons that exhibit a linear dichroism and are respectively bright under excitation by linearly polarized light in the x and y directions. Their energy difference will disappear in the absence of electron-hole exchange interaction, as shown in Fig. R2b. The reason for this exchange-driven splitting of bright states is discussed in greater detail in our response to the following comment. We have added both Fig. R1 and R2 to the SI and added the exciton binding energy to the main text.

Figure R2. Spectrum of exciton eigen-energy levels for S-NPB, calculated with electron-hole exchange energy in **a** and without electron-hole exchange energy in **b**. In the figures, b1 and b1 stand for linearly polarized light with polarization in the $\Gamma - X$ (b1) and $\Gamma - Y$ (b2) directions, while L and R stand for LCP and RCP. The lines are colored according to the oscillator strength of each exciton state with the given polarizations of light. Bright states are red, and dark states are blue. Note that the bright in-plane excitons are split in panel **a**, but their splitting disappears in panel **b** once the exchange interaction is removed.

Comment 2

The effect of electron-hole exchange on the exciton fine structure has been calculated in model 2D layered lead bromides: With single monolayer PbBr_4 octahedra layers the computed bright-dark splitting in GW-BSE is ~ 30 meV (see Fig. 2 in *Adv. Optical Mater.* 2023, 2202801); experiments have shown ~ 28 meV (Dyksik *et al.*, *Sci. Adv.* 7, eabk0904, 2021) with minimal splitting of the in-plane polarized exciton fine structure levels.

So, it seems unlikely that electron-hole exchange, per se, could be responsible for the claimed 17meV in-plane polarized bright exciton splitting. Could the source of the splitting be the Rashba/Dresselhaus splitting? As the authors correctly claim, the R/D term that corresponds to the spin splitting and spin texture shown in Fig 3d does not split the transitions from the $\pm K_r$ valleys in an independent particle picture. However, it can also be shown that this R/D term does not split the bright in-plane polarized exciton, even with electron-hole exchange. This can be shown by analysis of the effect of the spin splitting on the exciton fine structure in 2D layered systems, including electron-hole exchange, within second order perturbation theory... which is almost certainly justified given the expected large exciton binding energy in monolayer thickness PbBr layered perovskites. So, it is not clear, to me, what is going on in this calculation. Few details are provided to enable disentangling.

We appreciate the Reviewer's point that typically one thinks of exchange interaction when discussing the splitting of the bright and dark exciton states, whereas the discussion of splitting of the bright states might require further analysis.

Regarding the model calculation by Quarti *et al.* in *Adv. Optical Mater.* **2023**, 2202801, when calculating the bright-dark exciton splitting (their Fig. 2), the authors introduced some significant simplifications. First, they adopt symmetrized atomistic models rather than the experimental structure, which means that there is no octahedral tilting (Fig. R3). This simplification gives the structure a tetragonal P4mmm space group symmetry. Second, they replaced the organic spacer with Cs atoms, which ignores the dielectric contrast between the organic and inorganic components. In our calculations, we only use the relaxed experimental structure, in which the octahedra are distorted and tilted. In Quarti *et al.*, the first four exciton states are composed of one dark state, two degenerate bright in-plane exciton states, and one bright out-of-plane exciton state. However, the two bright in-plane excitons are not degenerate once the tilting of the real crystal structure is taken into account.

Figure R3. The difference between atomic structure in literature and our work (a) Structure in the literature (*Adv. Optical Mater.* 2023, 2202801 Fig 1), (b) Structure in our work. Only the inorganic part is shown for visual clarity.

As discussed in our main text, we find a 17 meV energy splitting between the two bright exciton states in the material we studied. This is fully consistent with the work of Quarti where they found a 10 meV splitting between the two in-plane bright exciton once octahedral tilting was included. The magnitude of the splitting is not expected to be identical, since we are studying different structures with different degrees of tilting. The splitting arising from octahedral tilting results in a linear dichroism, with the lower-energy exciton state coupled to light polarized along the x axis and higher-energy exciton state coupled to the y axis. To validate our theoretical calculation in this revision, we experimentally measured the linearly polarized absorption (Fig. R4). Consistent with theory, we find a 13-meV energy shift of the absorption onset when the direction of the linearly polarized light is changed from x to y .

Now that we have established the splitting of the bright states, the second question pertains to the relationship between the exchange interaction and the splitting of the bright exciton states. The observation of the relationship between the exchange interaction and the 17-meV splitting of the bright in-plane exciton states is simply an empirical one. When the electron-hole interaction Hamiltonian includes exchange, the 17-meV splitting appears; when the exchange is not included, the splitting is reduced to 1 meV. This is shown in Fig. S9, where we directly calculated the ϵ'' ($\Delta\epsilon''$) spectrum with and without the exchange interaction.

Figure R4 Reflectance spectra measured with linearly polarized light along x and y direction in **a**, and the fitted permittivity along these two directions with center wavelengths indicated at the top of the panels.

The inorganic part of our relaxed experimental structure belongs to the chiral $P2_1$ space group, which is a subgroup of the $P2_1/c$ analyzed in the Quarti paper because of the existence of stronger distortion. Therefore, the insights from their analysis are also applicable to our structure. The twice degenerate in-plane exciton splits due to the absence of irreducible representations of dimension larger than one. Our investigation extends their findings, showing that both the symmetry decrease and the electron-hole exchange energy contribute to the energy splitting of the bright in-plane states. Essentially, in the absence of exchange, the bright in-plane exciton states are only split by 1 meV. The exchange interaction results in the usual splitting of the spin-forbidden dark states and the spin-allowed bright states. However, due to their asymmetry, the two bright exciton states are not split from the dark states by the same amount. This results in a splitting of the bright states, which is absent if the exchange is turned off. We have modified the discussion in the main text to clarify this point as follows.

“Fig. 3a shows the theoretically calculated ϵ'' (and ϵ' obtained from the KK relation) overlaid on the experimental curves, and Fig. 3b shows $\Delta\epsilon''$ and $\Delta\epsilon'$ between LCP and RCP, all for S -NBP. The theoretical results are in excellent agreement with experimental spectral features, including the derivative-like lineshape and sign changes in the $\Delta\epsilon''$ spectrum. The binding energy of the lowest energy bright exciton of 0.46 eV is consistent with previously reported exciton binding energies in layered perovskites.⁴⁶ Importantly, we find that the lowest energy peak in ϵ'' consists of two distinct bright exciton resonances that are split by 17 meV. The lower-energy state is bright when the light is polarized along the Γ to X direction, and the higher-energy state is bright when the light is polarized along the Γ to Y direction (see Supplementary Fig. 10). Both states are bright under excitation with circularly polarized light. These two bright excitons are composed primarily of electrons in the two lowest conduction bands and holes in the two highest valence bands (see Supplementary Fig. 11f). Atomic-orbital-resolved band structures (Fig. 3c) and spin-resolved band structures (Fig. 3d) reveal that the highest valence bands and lowest conduction bands are RD-split states resulting from the Pb-Br octahedral tilting coupled with the large spin-orbit coupling (SOC) of Pb.

In an independent-particle picture, we would expect such states to give rise to two degenerate band-to-band transitions, one of which is allowed for RCP, and the other allowed for LCP due to selection rules imposed by the spin texture. Due to the degeneracy of these two sets of transitions, contrary to previous hypotheses that the RD splitting alone is responsible for chirality transfer between sublattices,⁴⁷ we would not expect to observe CD (though CPL is still possible). However, the degeneracy of these transitions is broken after accounting for strong exciton effects in 2D perovskites.⁴⁸ As established in previous work^{6,49}, the undistorted layered perovskite has two degenerate low-energy spin-allowed exciton states that are bright under excitation with in-plane-polarized light. The octahedral tilting breaks the degeneracy of these bright excitons, and the two states exhibit a linear dichroism with one state coupling to light polarized along X direction and the other coupling to light polarized along the Y direction. In the absence of the exchange interaction, these bright spin-allowed states are degenerate with their spin-forbidden counterparts, and the states themselves are nearly degenerate with a small crystal-field splitting of ~1 meV. However, this near degeneracy is lifted by electron-hole exchange interaction, which splits the two sets of bright and dark states by different amounts, resulting in a ~17 meV splitting of the two bright states (Supplementary Note 4). This splitting of the bright states, which couple dissimilarly to RCP and LCP, gives rise to a pronounced derivative-like line-shape in the $\Delta\epsilon''$ spectrum, reminiscent of the Cotton effect.⁵⁰ In the absence of exchange, the excitonic states are nearly degenerate and the CD coming from band-edge excitons vanishes (Supplementary Fig. 9). Thus, the large chirality transfer from chiral spacers to lead-bromide sublattice, and the observed Cotton effect, are quantum phenomena driven by the different exchange-scattering of the bright excitons, which is in turn related to the crystal field splitting.”

Comment 3

We can however examine the calculation from the standpoint of what went into it with respect to symmetry. The Authors claim that the necessary conditions to observe CD (from SI page S18). Missing from this list is anything that requires chirality. All of the “necessary conditions” listed by the Authors can occur in non-chiral systems.

For example, the Authors emphasize the importance of the RD splitting shown in Fig. 3d. This R/D type spin splitting is due to bulk inversion asymmetry which creates linear-in-K spin splitting according to the invariant, (see Jana *et al.*, *Nat. Commun.* 2021, 12:4982), $H_{BIA} = \alpha_{zx} S_z K_x$. Here, S_z and K_x are spin operators and wavevector respectively along the Z and X directions. This invariant gives spin polarization in the out-of-plane (z) direction for wavevector in the in-plane X direction, normal to the 2₋₁ screw axis, which is what the authors show in Fig. 3d.

However, this R/D term exists in non-chiral systems, as does electron-hole exchange obviously. So, there is something critical missing from the Authors’ list of “necessary conditions”. Again: nothing that the authors list above as critical ingredients requires *chirality*, and yet we know that intrinsic CD requires chirality. Consequently, either there is physics going on in the GW-BSE calculation that the Authors have not conveyed, or there is some other issue.

This was indeed an oversight on our part. What we meant to write is that the features seen in the CD, that is the appearance of the Cotton effect at the inorganic band edge, comes from the exchange interaction, not that CD in general comes from the exchange interaction. The unwritten additional condition for the appearance of CD in general is, as the Reviewer noted, the fact that the crystal must belong to a chiral space group. The inorganic part of our relaxed experimental structure belongs to the $P2_1$ space group, which is indeed a chiral space group. The relationship between Rashba splitting, exchange, and the Cotton effect that we observe is noteworthy because it leads to an exceptional enhancement of the CD at the band edge, where the electronic states arise from the inorganic sublattice.

We have revised the manuscript and the SI to clarify this point. Specifically, we have modified the main text to read “Theoretical calculations show that the Cotton effect seen in the CD spectrum is an excitonic effect driven by an electron-hole exchange interaction between RD-split states at the band edge”.

Furthermore, we have modified the SI to read “We list several necessary conditions for the formation of the exchange-driven Cotton effect: 1) a breaking of the inversion symmetry; 2) strong spin-orbit coupling; 3) the first few conduction bands and valence bands are isolated RD-split bands; 4) a strongly bound and isolated 1st exciton peak; 5) a finite exchange energy among the relevant exciton states; 6) an overall chiral space group.”

Comment 4

In SI Note 4 the authors provide a simple analysis based on Fermi’s golden rule for the circular polarization dependent permittivity. They arrive at an equation for the CD as follows:

$$\Delta\varepsilon''(\omega) = \frac{16\pi^2 e^2}{\omega^2} \sum_{vck} \text{Im}(X_E^x X_M^{y*} - X_E^y X_M^{x*})_{vck} \delta(\omega - \omega_{vck})$$

where X_E^i and X_M^i are components of the electric and magnetic transition dipole matrix elements. This looks to me as though the Authors are saying that $\text{CD} \sim \text{Im}[\vec{X}_E \times \vec{X}_m]$, i.e., that the CD should go like the cross product of the electric and magnetic transition dipoles. If this expression was actually used in the computations reported in the manuscript, I do not know what they are calculating. Unless I am misunderstanding something in a big way, the given equation is incorrect. It is well known that CD goes as the inner product of the electric and magnetic transition matrix elements, $\text{CD} \sim \text{Im}[\vec{X}_E \times \vec{X}_m]$, see Andrews and Tretton, J. Chem. Educ. 2020, 97, 4370-4376, where a similar FGR analysis leading to the expression for the CD response given and which shows this quite clearly.

We apologize that our notation was not clear here. We indeed calculated the CD in the same way as in Andrews and Tretton. In our notation, the magnetic and electric dipole terms are scalar values, not vectors, since we have already taken a dot product with the vector field of the light. The superscript on both the magnetic and electric terms refers to the direction of polarization of the

light. Hence, the actual magnetic field is oriented perpendicular to the polarization of light. In the paper by Andrews and Tretton, the CD for randomly oriented molecules is expressed as $\epsilon_{CD} = \frac{4k}{3c} \text{Im}(\vec{\mu}_{fi} \cdot \vec{m}_{if})$, where $\vec{\mu}_{fi}$ represents the electric dipole moment and \vec{m}_{if} denotes the magnetic dipole moment of the molecule. However, our definition for X_E and X_M are not identical to $\vec{\mu}_{fi}$ and \vec{m}_{if} . Here, we show that these two notation systems are equivalent when computing the CD for molecules with random orientations.

First, our definition for the $\hat{\epsilon}$ is the direction of the vector potential: $\hat{\epsilon} = \mathbf{A}/|\mathbf{A}|$. Assume that $\mathbf{A} = A_x \hat{x} + A_y \hat{y}$ and $|\mathbf{A}|$ is a constant, then

$$\nabla \times \hat{\epsilon} = \frac{-\frac{\partial}{\partial z} A_y \hat{x} + \frac{\partial}{\partial z} A_x \hat{y}}{|\mathbf{A}|}$$

Therefore,

$$X_E^{vck} = \hat{\epsilon} \cdot \langle v\mathbf{k} | \hat{\mathbf{p}} | c\mathbf{k} \rangle = \frac{\langle v\mathbf{k} | \hat{p}_x | c\mathbf{k} \rangle A_x + \langle v\mathbf{k} | \hat{p}_y | c\mathbf{k} \rangle A_y}{|\mathbf{A}|}$$

$$X_M^{vck} = \frac{1}{2} \langle v\mathbf{k} | \hat{\mathbf{L}} \cdot (\nabla \times \hat{\epsilon}) | c\mathbf{k} \rangle = \frac{-\frac{1}{2} \langle v\mathbf{k} | \hat{L}_x | c\mathbf{k} \rangle \cdot \frac{\partial}{\partial z} A_y + \frac{1}{2} \langle v\mathbf{k} | \hat{L}_y | c\mathbf{k} \rangle \cdot \frac{\partial}{\partial z} A_x}{|\mathbf{A}|}$$

Here, X_E^{vck} and X_M^{vck} are two scalars. In the expression used in the Supplementary Note 4, the superscript corresponds to the case when the vector potential is along a particular direction:

$$(X_E^x)_{vck} = \frac{\langle v\mathbf{k} | \hat{p}_x | c\mathbf{k} \rangle A_x}{|\mathbf{A}|}$$

$$(X_E^y)_{vck} = \frac{\langle v\mathbf{k} | \hat{p}_y | c\mathbf{k} \rangle A_y}{|\mathbf{A}|}$$

$$(X_M^x)_{vck} = \frac{1}{2} \langle v\mathbf{k} | \hat{L}_y | c\mathbf{k} \rangle \frac{\partial A_x}{\partial z} / |\mathbf{A}|$$

$$(X_M^y)_{vck} = -\frac{1}{2} \langle v\mathbf{k} | \hat{L}_x | c\mathbf{k} \rangle \frac{\partial A_y}{\partial z} / |\mathbf{A}|$$

Therefore, $X_M^x \propto \langle v\mathbf{k} | \hat{L}_y | c\mathbf{k} \rangle$ and $X_M^y \propto \langle v\mathbf{k} | \hat{L}_x | c\mathbf{k} \rangle$.

One can see that \vec{X}_E and \vec{X}_M are not vectors, since $X_E^{vck} = \hat{\epsilon} \cdot \langle v\mathbf{k} | \hat{\mathbf{p}} | c\mathbf{k} \rangle$ and $X_M^{vck} = \frac{1}{2} \langle v\mathbf{k} | \hat{\mathbf{L}} \cdot (\nabla \times \hat{\epsilon}) | c\mathbf{k} \rangle$ are both scalars. X_M^x and X_M^y are the scalars calculated for particular directions of the vector potential and are not identical to the magnetic dipoles $\vec{m}_{if,x}$ and $\vec{m}_{if,y}$.

As a result, CD in the z direction is:

$$CD_z \propto \text{Im}(X_E^x X_M^{y*} - X_E^y X_M^{x*})_{vck} \propto \text{Im}(p_x L_x^* + p_y L_y^*)_{vck}$$

where p_x , p_y , L_x and L_y stand for electron momentum and electron angular momentum in the x and y directions. CD for molecules with random orientations is the average CD over all three directions, i.e.,

$$CD = \frac{CD_x + CD_y + CD_z}{3} \propto \text{Im}(p_x L_x^* + p_y L_y^* + p_z L_z^*)$$

which agrees with the well-known inner product expression for CD of molecules.

To improve the clarity of our presentation, we have revised the Supplementary Note 4 to define all the expressions more clearly.

Comment 5

The computation of the magnetic dipole transition matrix element is known to be challenging. The Authors in Note 4 explain how they calculate it: they discard the spin portion and retain the orbital angular momentum portion. The expression that they give for the magnetic dipole transition matrix element is given on SI page S14

We can then apply a sum-over-states method to calculate the angular momentum matrix elements from the momentum matrix elements.^{13, 14, 15}

$$L_x(\mathbf{k}; m, n) = \frac{i\hbar}{m_e} \sum_{n'} \left(\frac{\langle \mathbf{k}m | p_y | \mathbf{k}n' \rangle \langle \mathbf{k}n' | p_z | \mathbf{k}m \rangle}{E_{m\mathbf{k}} - E_{n'\mathbf{k}}} - \frac{\langle \mathbf{k}m | p_z | \mathbf{k}n' \rangle \langle \mathbf{k}n' | p_y | \mathbf{k}m \rangle}{E_{m\mathbf{k}} - E_{n'\mathbf{k}}} \right).$$

L_y and L_z can be calculated in a similar fashion.

The problem is that this expression gives something proportional to the orbital magnetic moment of a given state, it is NOT a transition matrix element. The left-hand side of the expression involves band indices m, n ; the right-hand side does not involve the index n . The equation is simply incorrect. All the references given (Refs 13-15 in the SI) pertain to the calculation of the Lande g -factor of carriers, NOT to magnetic dipole transition matrix elements. So, if the given expression was actually used in the computations reported in this manuscript, I do not know what they are calculating, but it is not recognizable to me as CD.

This was a typo, and we thank the Reviewer for catching this. The correct expression should be

$$L_x(\mathbf{k}; m, n) = \frac{i\hbar}{m_e} \sum_{n'} \left(\frac{\langle \mathbf{k}m | p_y | \mathbf{k}n' \rangle \langle \mathbf{k}n' | p_z | \mathbf{k}n \rangle}{E_{mk} - E_{n'k}} - \frac{\langle \mathbf{k}m | p_z | \mathbf{k}n' \rangle \langle \mathbf{k}n' | p_y | \mathbf{k}n \rangle}{E_{mk} - E_{n'k}} \right)$$

We made this correction in the revised Supplementary Note 4. Note that we cited the previous work on the Lande g-factor, not because we use the identical expression, but because it was a nice reference for the derivation of the sum-over-states approach.

Comment 6

Minor issues:

a. Here and there in the Authors' mathematical expressions the term $(\nabla \times \hat{\epsilon})$ appears. For example on Page S13: $X_M^{vck} = \frac{1}{2} \langle v\mathbf{k} | \hat{\mathbf{L}} \cdot (\nabla \times \hat{\epsilon}) | c\mathbf{k} \rangle$. Since $\hat{\epsilon}$ is defined in Note 4 as a unit vector, this term vanishes. Such expressions are missing something, and this should be fixed.

b. Different definitions are used for right and left polarization vectors. For example, in Note 4 the definition is given $\hat{\epsilon}_L = (\hat{x} + i\hat{y})/\sqrt{2}$, which is correct according to IUPAC convention. But in Note 3 the opposite definition is given: $\hat{\epsilon}_L = (\hat{x} - i\hat{y})/\sqrt{2}$. This sort of inconsistency should be corrected.

To clarify point a, we use $\hat{\epsilon} = \mathbf{A}/|\mathbf{A}|$. Therefore, $\nabla \times \hat{\epsilon} = \frac{-\frac{\partial}{\partial z} A_y \hat{x} + \frac{\partial}{\partial z} A_x \hat{y}}{|\mathbf{A}|}$ is nonzero. For point b, we have updated the notation used in Supplementary Note 3 to ensure consistency between theory and experiments.

Comment 7

On page 9 the Authors write: "... contrary to previous hypotheses that the RD splitting alone is responsible for chirality transfer between sublattices [24, 46] we would not expect to observe CD". I think that the Authors here mean that RD splitting cannot be responsible for CD. Ref 24 made no such claim (although Ref 46 did... and the claim is not correct). Ref 24 is a study on the mechanism of chirality transfer from the chiral organics to the inorganic sublattice – which results in RD splitting of bands associated with the inorganic sublattice. Ref 24 is agnostic about the mechanism of CD.

Yes, the Reviewer is correct that we mean that RD-splitting cannot be responsible for CD. We have removed the citation of Ref. 24.

REVIEWER COMMENTS

Reviewer #2 (Remarks to the Author):

The manuscript has been properly revised according to the reviewers' comment. I now have no further question.

Reviewer #3 (Remarks to the Author):

Please see attached referee report

Referee Report: Nature Communications manuscript 23-39273A

Title: Large exchange-driven intrinsic circular dichroism of a chiral 2D hybrid perovskite

Authors:

Shunran Li, Xian Xu, Conrad A. Kocoj, Chenyu Zhou, Yanyan Li, Du Chen, Joseph A. Bennett, Sunhao Liu, Lina Quan, Suchismita Sarker, Mingzhao Liu, Diana Y. Qiu, Peijun Guo

Executive summary and recommendations:

This revised manuscript is substantially improved from the originally submitted version. Additional experimental data pertaining to structure of thin films versus single crystals has been provided; and requested control measurements have been done on thin film samples. Most of the issues that I previously noted with equations in original submission have been corrected.

The claim made in the current manuscript is that the single crystals show $\sim 10x$ larger CD than thin films. That is reasonable especially given the new structural data added in the revised manuscript.

However: I still suspect that there is some problem with the conclusions claimed by the Authors regarding the *magnitude* of the CD inferred from the single crystals. *The problem* is that the g_{CD} values for the exciton are in the range of ± 0.1 (see Supp Fig 12), which is enormous and it seems much too large: The maximum possible value of $g_{CD} = CD/A = \Delta A/A$, where A is the decadic absorbance and $CD = \Delta A$ is the difference in decadic absorbance between left and right polarized light, can be estimated to go as $g_{CD} \sim 2 \hbar k / |P|$, where k is the wave vector of light absorbed at the exciton resonance and $|P|$ is the magnitude of the Kane momentum matrix element at the band edge (see for example 10.1103/PhysRevB.86.205301 for a definition of the Kane matrix element if this is unfamiliar) assuming that the magnitude of the orbital component of the magnetic transition dipole is of order \hbar when evaluated between the band edge Bloch states of the conduction and valence bands (which is likely an overestimate). Using the refractive index shown in Fig 2c, and estimating the Kane energy $E_p = 2|P|^2/m_e$ of the order 10 eV which is reasonable for perovskites, and the wavelength of 390 nm this estimate would give maximum g_{CD} of ~ 0.004 in the *narrow linewidth limit* where the bright in-plane exciton transitions are spectrally isolated. This is ~ 25 times smaller than what the Authors are reporting here and the bright exciton transitions are not spectrally isolated (the broadening used in these calculations is given as 46 meV on page 7 of the Authors' rebuttal letter, which is larger than the in-plane polarizes bright exciton splitting calculated by the Authors).

Nevertheless, despite this reservation, **I recommend publication of the manuscript with minor revisions**. My estimate for the maximal g_{CD} above may be wrong; and I agree with the Authors when they write in the rebuttal letter that: the “present manuscript provides a first experimental demonstration of intrinsic CD extraction for single crystals, illustrates the difference between crystals and films, and offers a full theoretical account of the intrinsic CD response of the chosen chiral 2D perovskite”. If there is an issue with the calculated magnitude of the CD dissymmetry, it will come out in discussions and further work within the community. Meantime, this work contains much exciting new data and much that is new in terms of both experimental and theoretical approaches and all of this should be made available to the community in order for that discussion to occur.

My detailed comments and recommendations for revisions follow.

Detailed comments:

- **On the experimental side**, the Authors have added a great deal of additional structural data and analysis pertaining to the claimed ~ 100 x difference in dissymmetry g_{cd} values between single crystal and thin film samples.
 - Synchrotron studies were added (Supp Fig 17) to provide further insight into the structural differences;
 - a control work-up of reflectivity data measured on thin film samples, following the same methodology as the Authors used in their analysis of the single crystal samples, was added as requested to substantiate the claimed differences in CD between single crystal and thin film samples.
 - The lack of difference in the thin film and single crystal data originally presented data in Fig 5 and Fig 4 of the original submission, which did not support the claimed ~ 100 x difference in dissymmetry g_{cd} (as acknowledged by the Authors in their rebuttal), is now explained in terms of the originally presented thin film data having been specifically selected for high CD values which have now been found to correspond to stripe-like domains. The revised presentation in the new Fig 4 e,f seems much more balanced/representation as it is accompanied by statistically information.

On the theoretical side the authors have largely clarified and/or corrected the equation issues noted in my previous review:

- They clarified the equation now in line 239 of the SI;
- They corrected the equation now on line 250 of the SI),
- They have explained what they mean when they write expressions such as $\nabla \times \hat{\epsilon}$ in lines 224 and 265 of the SI.
- More important the Authors have added a detailed discussion and additional figures bearing on the computational details (Suppl Fig 11) and specifically on the exciton fine structure and optical properties of the fine structure levels (Suppl Fig. 10). This data is wonderful and very useful.

Suggested revisions:

- The Authors now explain what they mean when they write expressions such as $\nabla \times \hat{\epsilon}$ in lines 224 and 265 of the SI, which was confusing since elsewhere $\hat{\epsilon}$ is used as a unit vector which has no spatial dependence. The usage in lines 224/265 is now defined in line 226 as $\hat{\epsilon} = \mathbf{A}/|\mathbf{A}|$. This notation is frankly still confusing: It is incompatible with

the notation in line 204 for example:

207
$$A = |A_0| \sum_k \hat{\epsilon} (e^{ik \cdot r} \hat{a}_k + e^{-ik \cdot r} \hat{a}_k^\dagger)$$

because in lines 224/265 etc. the quantity $\hat{\epsilon}$ has a spatial dependence while in line 207 it clearly does NOT. The Authors can do as they see fit, but I suggest that they could write something like $\hat{A} = \mathbf{A}/|\mathbf{A}|$ in lines 224 and 265 of the SI that would be more transparent and less confusing to readers.

- I suggest that the authors add equation numbers in the SI. For example, refer to the paragraph just written above. Having to refer to specific equations by the line numbers at which they appear in the manuscript is a hindrance to communication. There is a reason it is conventional in complex mathematical analysis to use equation numbers, the Authors would help their readers by following this convention.
- The juxtaposition of single crystal data and thin film data in the new Figure 4 seems to invite comparison of the CD of single crystals and thin film samples (panel (d) versus panels (e,f)), but the panels that provide for this comparison show different quantities and cannot be directly compared...can the Authors provide a comparison of like quantities here? I do realize that a direct comparison is available between Fig 2a and fig 4e,f.
- In some places the figure panels and the figure captions do not match. For example, the caption for Fig 4 (d) states “**d** Ratios in the voltages of the 1f and the DC components measured on S-NPB at 410 various temperatures from 290 K to 50 K” but what is plotted in the panel is NOT the voltage ratio but the calculated ellipticity:

- These quantities are related (see lines 275-277 in the manuscript) but they are NOT the same thing. This is confusing and should be corrected; other figures should be double-checked for like issues.
- The authors state in the rebuttal letter that “a constant broadening of 46 meV was included as an experiment-informed empirical parameter”. The value of g_{CD} will depend on the broadening; the value of the broadening should be clearly stated in the text somewhere where readers can find it easily.

Reviewer #1

This revised manuscript is substantially improved from the originally submitted version. Additional experimental data pertaining to structure of thin films versus single crystals has been provided; and requested control measurements have been done on thin film samples. Most of the issues that I previously noted with equations in original submission have been corrected.

The claim made in the current manuscript is that the single crystals show $\sim 10x$ larger CD than thin films. This is reasonable especially given the new structural data added in the revised manuscript.

However, I still suspect that there is some problem with the conclusions claimed by the Authors regarding the magnitude of the CD inferred from the single crystals. The problem is that the g_{CD} values for the exciton are in the range of ± 0.1 (see Supp Fig 12), which is enormous and it seems much too large: The maximum possible value of $g_{CD} = CD/A = \Delta A/A$, where A is the decadic absorbance and $CD = \Delta A$ is the difference in decadic absorbance between left and right polarized light, can be estimated to go as $g_{CD} \sim 2\hbar k/|P|$, where k is the wave vector of light absorbed at the exciton resonance and $|P|$ is the magnitude of the Kane momentum matrix element at the band edge (see for example 10.1103/PhysRevB.86.205301 for a definition of the Kane matrix element if this is unfamiliar) assuming that the magnitude of the orbital component of the magnetic transition dipole is of order \hbar when evaluated between the band edge Bloch states of the conduction and valence bands (which is likely an overestimate). Using the refractive index shown in Fig 2c, and estimating the Kane energy $E_P = 2|P|^2/m_e$ of the order 10 eV which is reasonable for perovskites, and the wavelength of 390 nm this estimate would give maximum g_{CD} of ~ 0.004 in the narrow linewidth limit where the bright in-plane exciton transitions are spectrally isolated. This is ~ 25 times smaller than what the Authors are reporting here and the bright exciton transitions are not spectrally isolated (the broadening used in these calculations is given as 46 me on page 7 of the Authors' rebuttal letter, which is larger than the in-plane polarized bright exciton splitting calculated by the Authors).

Nevertheless, despite this reservation, I recommend publication of the manuscript with minor revisions. My estimate for the maximal g_{CD} above may be wrong; and I agree with the Authors when they write in the rebuttal letter that: the “present manuscript provides a first experimental demonstration of intrinsic CD extraction for single crystals, illustrates the difference between crystals and films, and offers a full theoretical account of the intrinsic CD response of the chosen chiral 2D perovskite”. If there is an issue with the calculated magnitude of the CD dissymmetry, it will come out in discussions and further work within the community. Meantime, this work contains much exciting new data and much that is new in terms of both experimental and theoretical approaches, and all of this should be made available to the community in order for that discussion to occur.

My detailed comments and recommendations for revisions follow.

Detailed comments:

On the experimental side, the Authors have added a great deal of additional structural data and analysis pertaining to the claimed ~ 100 x difference in dissymmetry g_{CD} values between single crystal and thin film samples.

Synchrotron studies were added (Supp Fig 17) to provide further insight into the structural differences.

A control work-up of reflectivity data measured on thin film samples, following the same methodology as the Authors used in their analysis of the single crystal samples, was added as requested to substantiate the claimed difference in CD between single crystal and thin film samples.

The lack of difference in the thin film and single crystal data originally presented in Fig 5 and Fig 4 of the original submission, which did not support the claimed ~ 100 x difference in dissymmetry g_{CD} (as acknowledged by the Authors in their rebuttal), is now explained in terms of the originally presented thin film data having been specifically selected for high CD values which have now been found to correspond to stripe-like domains. The revised presentation in the new Fig 4e,f seems much more balanced/representation as it is accompanied by statistical information.

On the theoretical side, the authors have largely clarified and/or corrected the equation issues noted in my previous review:

They clarified the equation now in line 239 of the SI.

They corrected the equation now on line 250 of the SI.

They have explained what they mean when they write expressions such as $\nabla \times \hat{\epsilon}$ in line 224 and 265 of the SI.

More importantly the Authors have added a detailed discussion and additional figure bearing on the computational details (Supp Fig 11) and specifically on the exciton fine structure and optical properties of the fine structure levels (Supp Fig. 10). This data is wonderful and very useful.

Our response: we thank the Reviewer for the positive feedback on our manuscript. We address the comments in detail below.

Suggested revisions:

1. The Authors now explain what they mean when they write expressions such as $\nabla \times \hat{\epsilon}$ in lines 224 and 265 of the SI, which was confusing since elsewhere $\hat{\epsilon}$ is used as a unit vector which has no spatial dependence. The usage in lines 224/265 is now defined in line 226 as $\hat{\epsilon} = \mathbf{A}/|\mathbf{A}|$. This notation is frankly still confusing: it is incompatible with the notation in line 204 for example:

207

$$\mathbf{A} = |\mathbf{A}_0| \sum_{\mathbf{k}} \hat{\epsilon} (e^{i\mathbf{k}\cdot\mathbf{r}} \hat{a}_{\mathbf{k}} + e^{-i\mathbf{k}\cdot\mathbf{r}} \hat{a}_{\mathbf{k}}^\dagger)$$

Because in line 224/265 etc. the quantity $\hat{\epsilon}$ has a spatial dependence while in line 207 it clearly does NOT. The Authors can do as they see fit, but I suggest that they could write something like $\hat{\mathbf{A}} = \mathbf{A}/|\mathbf{A}|$ in lines 224 and 265 of the SI that would be more transparent and less confusing to readers.

We have made this update as suggested by the Reviewer.

2. I suggest that the authors add equation numbers in the SI. For example, refer to the paragraph just written above. Having to refer the specific equations by the line numbers at which they appear in the manuscript is a hindrance to communication. There is a reason it is conventional in complex mathematical analysis to use equation numbers; the Authors would help their readers by following this convention.

We have made this update as suggested by the Reviewer. All the important equations in the SI are now labeled properly.

3. The juxtaposition of single crystal data and thin film data in the new Figure 4 seems to invite comparison of the CD of single crystals and thin film samples (panel d verses panel e,f, but the panels that provide for this comparison show different quantities and cannot be directly compared ... can the Authors provide a comparison of like quantities here? I do realize that a direct comparison is available between Fig 2a and Fig 4e,f.

We thank the Reviewer for this nuanced suggestion. In the revised manuscript, we have plotted the reflectance ellipticity θ in replacement of the V_{1f}/V_{DC} that appeared as Fig. 4f. This allows a direct comparison between Fig. 4d and Fig. 4f.

4. In some places the figure panels and the figure captions do not match. For example, the caption for Fig 4d states “d Ratios in the voltages of the 1f and the DC components measured on S-NPB at various temperatures from 290 K to 50 K” but what is plotted in the panel is NOT the voltage ratio but the calculated ellipticity. These quantities are related (see lines 275-277 in the manuscript) but they are NOT the same thing. This is confusing and should be corrected; other figures should be double-checked for like issues.

We have rectified this in the revised manuscript. We also double-checked other sections of the manuscript to correct any remaining typographical errors.

5. The authors state in the rebuttal letter that “a constant broadening of 46 meV was included as an experiment-informed empirical parameter”. The value of g_{CD} will depend on the broadening; the value of the broadening should be clearly stated in the text somewhere where readers can find it easily.

This has now been included in the main text, which now reads “With a constant broadening of 46 meV, the theoretical results are in excellent agreement with experimental spectral features...” We also clarified this point in the caption for Fig. 3a.

REVIEWERS' COMMENTS

Reviewer #3 (Remarks to the Author):

The Authors have responded adequately/appropriately to my comments/ questions and have made appropriate revisions.

I believe that this work contains much exciting new data and new theoretical results which will benefit the community and spur further development in the field.

I recommend that the work be published in its current form.

Comments in detail - for information only- are included in the attached report.

Referee Report: Nature Communications manuscript NCOMMS-23-39273C

Title: Large exchange-driven intrinsic circular dichroism of a chiral 2D hybrid perovskite

Authors:

Shunran Li, Xian Xu, Conrad A. Kocoj, Chenyu Zhou, Yanyan Li, Du Chen, Joseph A. Bennett, Sunhao Liu, Lina Quan, Suchismita Sarker, Mingzhao Liu, Diana Y. Qiu, Peijun Guo

The Authors have responded adequately/appropriately to my comments/ questions and have made appropriate revisions.

I believe that this work contains much exciting new data and new theoretical results which will benefit the community and spur further work.

I recommend that the work be published in its current form.

Detailed comments (for information only, I do not need or want to see this manuscript for review again):

1. Regarding the question I raised about the magnitude of the anisotropy factor (g_{CD}) that they report: I agree with the Authors' response that,

Thus, while the Reviewer raises a valuable point, we feel it falls beyond the current scope of this work, and as the reviewer states resolving this question can be left to "discussions and further work within the community."

I will point out to the Authors, just for information, that the Kane energy parameter is tightly constrained by measurements (or first-principles calculations) of the band-edge effective masses, Lande g-factors, radiative transition rates etc., which are all influenced strongly by the Kane matrix element: It is therefore virtually certain that the estimate they point to in their response (namely, $E_p \sim 120$ eV from 10.1103/PhysRevMaterials.3.111601) cannot be correct. In any event in order to resolve the discrepancy here at issue, the Kane energy would need to be *much* smaller than my estimate of 10 eV, not larger. But both the Authors and I agree that this issue must remain unresolved pending future study.

2. The suggested revisions pertaining to nomenclature of unit vectors versus the dimensionless polarization vector have been implemented- Thank you Authors!
3. The Authors accepted my suggestion of adding Eq. numbers in the SI to help the reader, - again, my thanks.
4. The Authors have revised Fig 4 panels d,f to facilitate direct comparison of reflectance ellipticity between single crystal and annealed thin film samples- This is a helpful addition--Thank you again.
5. Various noted typos and inconsistencies between captions and figure panels have been corrected.
6. The Authors have added the critical information of the value of the empirical broadening parameter, which was previously missing, in the main text and in the caption of Fig 3. This is a good and important addition.